# Neural (Tangent Kernel) Collapse

**Mariia Seleznova**[1]* **Dana Weitzner**[2] **Raja Giryes**[2] **Gitta Kutyniok**[1] **Hung-Hsu Chou**[1]
[1]Ludwig-Maximilians-Universität München [2]Tel Aviv University

## Abstract

This work bridges two important concepts: the Neural Tangent Kernel (NTK), which captures the evolution of deep neural networks (DNNs) during training, and the Neural Collapse (NC) phenomenon, which refers to the emergence of symmetry and structure in the last-layer features of well-trained classification DNNs. We adopt the natural assumption that the empirical NTK develops a block structure aligned with the class labels, i.e., samples within the same class have stronger correlations than samples from different classes. Under this assumption, we derive the dynamics of DNNs trained with mean squared (MSE) loss and break them into interpretable phases. Moreover, we identify an invariant that captures the essence of the dynamics, and use it to prove the emergence of NC in DNNs with block-structured NTK. We provide large-scale numerical experiments on three common DNN architectures and three benchmark datasets to support our theory.

## 1 Introduction

Deep Neural Networks (DNNs) are advancing the state of the art in many real-life applications, ranging from image classification to machine translation. Yet, there is no comprehensive theory that can explain a multitude of empirical phenomena observed in DNNs. In this work, we provide a theoretical connection between two such empirical phenomena, prominent in modern DNNs: *Neural Collapse (NC)* and *Neural Tangent Kernel (NTK) alignment*.

**Neural Collapse.** NC [39] emerges while training modern classification DNNs past zero error to further minimize the loss. During NC, the class means of the DNN's last-layer features form a symmetric structure with maximal separation angle, while the features of each individual sample collapse to their class means. This simple structure of the feature vectors appears favourable for generalization and robustness in the literature [12, 31, 40, 47]. Though NC is common in modern DNNs, explaining the mechanisms behind its emergence is challenging, since the complex non-linear training dynamics of DNNs evade analytical treatment.

**Neural Tangent Kernel.** The NTK [30] describes the gradient descent dynamics of DNNs in the function space, which provides a dual perspective to DNNs' evolution in the parameters space. This perspective allows to study the dynamics of DNNs analytically in the infinite-width limit, where the NTK is constant during training [30]. Hence, theoretical works often rely on the infinite-width NTK to analyze generalization of DNNs [1, 20, 28, 49]. However, multiple authors have argued that the infinite-width limit does not fully reflect the behaviour of realistic DNNs [2, 10, 22, 27, 36, 43], since constant NTK implies that no feature learning occurs during DNNs training.

**NTK Alignment.** While the infinite-width NTK is label-agnostic and does not change during training, the empirical NTK rapidly aligns with the target function in the early stages of training [5, 7, 44, 45]. In the context of classification, this manifests itself as the emergence of a block structure

---

*Correspondence to: Mariia Seleznova (`selez@math.lmu.de`).

37th Conference on Neural Information Processing Systems (NeurIPS 2023).

in the kernel matrix, where the correlations between samples from the same class are stronger than between samples from different classes. The NTK alignment implies the so-called local elasticity of DNNs' training dynamics, i.e., samples from one class have little impact on samples from other classes in Stochastic Gradient Descent (SGD) updates [23]. Several recent works have also linked the local elasticity of training dynamics to the emergence of NC [33, 53]. This brings us to the main question of this paper: *Is there a connection between NTK alignment and neural collapse?*

**Contribution.** In this work, we consider a model of NTK alignment, where the kernel has a *block structure*, i.e., it takes only three distinct values: an inter-class value, an intra-class value and a diagonal value. We describe this model in Section 3. Within the model, we establish the connection between NTK alignment and NC, and identify the conditions under which NC occurs. Our main contributions are as follows:

- We derive and analyze the training dynamics of DNNs with MSE loss and block-structured NTK in Section 4. We identify three distinct convergence rates in the dynamics, which correspond to three components of the training error: error of the global mean, of the class means, and of each individual sample. These components play a key role in the dynamics.

- We show that NC emerges in DNNs with block-structured NTK under additional assumptions in Section 5.3. To the best of our knowledge, this is the first work to connect NTK alignment and NC. While previous contributions rely on the unconstrained features models [21, 38, 48] or other imitations of DNNs' training dynamics [53] to derive NC (see Appendix A for a detailed discussion of related works), we consider standard gradient flow dynamics of DNNs simplified by our assumption on the NTK structure.

- We analyze when NC does or does not occur in DNNs with NTK alignment, both theoretically and empirically. In particular, we identify an invariant of the training dynamics that provides a necessary condition for the emergence of NC in Section 5.2. Since DNNs with block-structured NTK do not always converge to NC, we conclude that NTK alignment is a more widespread phenomenon than NC.

- We support our theory with large-scale numerical experiments in Section 6. Source code to reproduce the results is available in the project's GitHub repository.

## 2 Preliminaries

We consider the classification problem with $C \in \mathbb{N}$ classes, where the goal is to build a classifier that returns a class label for any input $x \in \mathcal{X}$. In this work, the classifier is a DNN trained on a dataset $\{(x_i, y_i)\}_{i=1}^N$, where $x_i \in \mathcal{X}$ are the inputs and $y_i \in \mathbb{R}^C$ are the one-hot encodings of the class labels. We view the output function of the DNN $f : \mathcal{X} \to \mathbb{R}^C$ as a composition of parametrized last-layer *features* $h : \mathcal{X} \to \mathbb{R}^n$ and a linear *classification* layer parametrized by weights $\mathbf{W} \in \mathbb{R}^{C \times n}$ and biases $\mathbf{b} \in \mathbb{R}^C$. Then the logits of the training data $X = \{x_i\}_{i=1}^N$ can be expressed as follows:

$$f(X) = \mathbf{W}\mathbf{H} + \mathbf{b}\mathbf{1}_N^\top, \tag{1}$$

where $\mathbf{H} \in \mathbb{R}^{n \times N}$ are the features of the entire dataset stacked as columns and $\mathbf{1}_N \in \mathbb{R}^N$ is a vector of ones. Though we omit the notion of the data dependence in the text to follow, i.e. we write $\mathbf{H}$ without the explicit dependence on $X$, we emphasize that the features $\mathbf{H}$ are a function of the data and the DNN's parameters, unlike in the previously studied unconstrained feature models [21, 38, 48].

We assume that the dataset is *balanced*, i.e. there are $m := N/C$ training samples for each class. Without loss of generality, we further assume that the inputs are reordered so that $x_{(c-1)m+1}, \ldots, x_{cm}$ belong to class $c$ for all $c \in [C]$. This will make the notation much easier later on. Since the dimension of features $n$ is typically much larger than the number of classes, we also assume $n > C$ in this work.

### 2.1 Neural Collapse

Neural Collapse (NC) is an empirical behaviour of classifier DNNs trained past zero error [39]. Let $\langle h \rangle := N^{-1} \sum_{i=1}^N h(x_i)$ denote the global features mean and $\langle h \rangle_c := m^{-1} \sum_{x_i \in \text{class } c} h(x_i)$, $c \in [C]$ be the class means. Furthermore, define the matrix of normalized centered class means as $\mathbf{M} := [\langle \overline{h} \rangle_1 / \|\langle \overline{h} \rangle_1\|_2, \ldots, \langle \overline{h} \rangle_C / \|\langle \overline{h} \rangle_C\|_2]^\top \in \mathbb{R}^{n \times C}$, where $\langle \overline{h} \rangle_c = \langle h \rangle_c - \langle h \rangle$, $c \in [C]$. We say that a DNN exhibits NC if the following four behaviours emerge as the training time $t$ increases:

(NC1) **Variability collapse:** for all samples $x_i^c$ from class $c \in [C]$, where $i \in [m]$, the penultimate layer features converge to their class means, i.e. $\|h(x_i^c) - \langle h \rangle_c\|_2 \to 0$.

(NC2) **Convergence to Simplex Equiangular Tight Frame (ETF):** for all $c, c' \in [C]$, the class means converge to the following configuration:

$$\|\langle h \rangle_c - \langle h \rangle\|_2 - \|\langle h \rangle_{c'} - \langle h \rangle\|_2 \to 0, \quad \mathbf{M}^\top \mathbf{M} \to \frac{C}{C-1}(\mathbb{I}_C - \frac{1}{C}\mathbf{1}_C\mathbf{1}_C^\top).$$

(NC3) **Convergence to self-duality:** the class means $\mathbf{M}$ and the final weights $\mathbf{W}^\top$ converge to each other:
$$\left\|\mathbf{M}/\|\mathbf{M}\|_F - \mathbf{W}^\top/\|\mathbf{W}^\top\|_F\right\|_F \to 0.$$

(NC4) **Simplification to Nearest Class Center (NCC):** the classifier converges to the NCC decision rule behaviour:

$$\operatorname*{argmax}_c(\mathbf{W}h(x) + \mathbf{b})_c \to \operatorname*{argmin}_c\|h(x) - \langle h \rangle_c\|_2.$$

Though NC is observed in practice, there is currently no conclusive theory on the mechanisms of its emergence during DNN training. Most theoretical works on NC adopt the unconstrained features model, where features $\mathbf{H}$ are free variables that can be directly optimized [21, 38, 48]. Training dynamics of such models do not accurately reflect the dynamics of real DNNs, since they ignore the dependence of the features on the input data and the DNN's trainable parameters. In this work, we make a step towards realistic DNN dynamics by means of the Neural Tangent Kernel (NTK).

## 2.2 Neural Tangent Kernel

The NTK $\Theta$ of a DNN with the output function $f : \mathcal{X} \to \mathbb{R}^C$ and trainable parameters $\mathbf{w} \in \mathbb{R}^P$ (stretched into a single vector) is given by

$$\Theta_{k,s}(x_i, x_j) := \langle \nabla_\mathbf{w} f_k(x_i), \nabla_\mathbf{w} f_s(x_j) \rangle, \quad x_i, x_j \in \mathcal{X}, \quad k, s \in [C]. \tag{2}$$

We also define the last-layer features kernel $\Theta^h$, which is a component of the NTK corresponding to the parameters up to the penultimate layer, as follows:

$$\Theta_{k,s}^h(x_i, x_j) := \langle \nabla_\mathbf{w} h_k(x_i), \nabla_\mathbf{w} h_s(x_j) \rangle, \quad x_i, x_j \in \mathcal{X}, \quad k, s \in [n]. \tag{3}$$

Intuitively, the NTK captures the correlations between the training samples in the DNN dynamics. While most theoretical works consider the infinite-width limit of DNNs [30, 52], where the NTK can be computed theoretically, empirical studies have also extensively explored the NTK of finite-width networks [19, 36, 45, 49]. Unlike the label-agnostic infinite-width NTK, the empirical NTK aligns with the labels during training. We use this observation in our main assumption (Section 3).

## 2.3 Classification with MSE Loss

We study NC for DNNs with the mean squared error (MSE) loss given by

$$\mathcal{L}(\mathbf{W}, \mathbf{H}, \mathbf{b}) = \frac{1}{2}\|f(X) - \mathbf{Y}\|_F^2, \tag{4}$$

where $\mathbf{Y} \in \mathbb{R}^{C \times N}$ is a matrix of stacked labels $y_i$. While NC was originally introduced for the cross-entropy (CE) loss [39], which is more common in classification problems, the MSE loss is much easier to analyze theoretically. Moreover, empirical observations suggest that DNNs with MSE loss achieve comparable performance to using CE [14, 29, 41], which motivates the recent line of research on MSE-NC [21, 38, 48].

## 3 Block Structure of the NTK

Numerous empirical studies have demonstrated that the NTK becomes aligned with the labels $\mathbf{Y}^\top\mathbf{Y}$ during the training process [7, 32, 45]. This alignment constitutes feature learning and is associated with better performance of DNNs [9, 13]. For classification problems, this means that the empirical

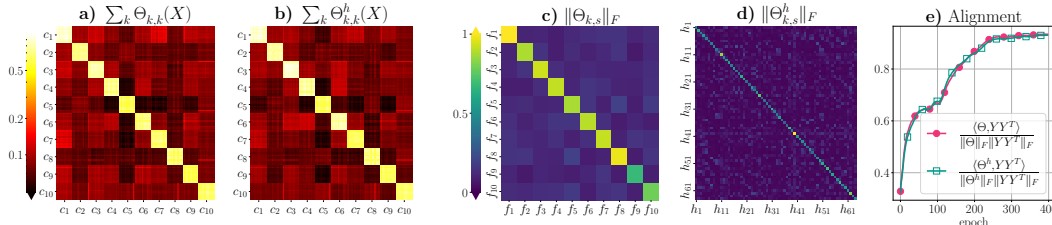

Figure 1: The NTK block structure of ResNet20 trained on MNIST. **a)** Traced kernel $\sum_{k=1}^{C} \Theta_{k,k}(X)$ computed on a random data subset with 12 samples from each class. The samples are ordered as described in Section 2, so that the diagonal blocks correspond to pairs of inputs from the same class. **b)** Traced kernel $\sum_{k=1}^{n} \Theta_{k,k}^{h}(X)$ computed on the same subset. **c)** Norms of the kernels $\Theta_{k,s}(X)$ for all $k, s \in [C]$. **d)** Norms of the kernels $\Theta_{k,s}^{h}(X)$ for all $k, s \in [n]$. The color bars show the values in each heatmap as a fraction of the maximal value in the heatmap. **e)** The alignment of the traced kernels from panes **a** and **b** with the class labels.

NTK develops an approximate block structure with larger kernel values corresponding to pairs of samples $(x_i^c, x_j^c)$ from the same class [44]. Figure 1 shows an example of such a structure emergent in the empirical NTK of ResNet20 trained on MNIST.[2] Motivated by these observations, we assume that the NTK and the last-layer features kernel exhibit a block structure, defined as follows:

**Definition 3.1** (Block structure of a kernel). *We say a kernel $\Theta : \mathcal{X} \times \mathcal{X} \to \mathbb{R}^{K \times K}$ has a block structure associated with $(\lambda_1, \lambda_2, \lambda_3)$, if $\lambda_1 > \lambda_2 > \lambda_3 \geq 0$ and*

$$\Theta(x,x) = \lambda_1 \mathbb{I}_K, \quad \Theta(x_i^c, x_j^c) = \lambda_2 \mathbb{I}_K, \quad \Theta(x_i^c, x_j^{c'}) = \lambda_3 \mathbb{I}_K, \tag{5}$$

*where $x_i^c$ and $x_j^c$ are two distinct inputs from the same class, and $x_j^{c'}$ is an input from class $c' \neq c$.*

**Assumption 3.2.** *The NTK $\Theta : \mathcal{X} \times \mathcal{X} \to \mathbb{R}^{C \times C}$ has a block structure associated with $(\gamma_d, \gamma_c, \gamma_n)$, and the penultimate kernel $\Theta^h : \mathcal{X} \times \mathcal{X} \to \mathbb{R}^{n \times n}$ has a block structure associated with $(\kappa_d, \kappa_c, \kappa_n)$.*

This assumption means that every kernel $\Theta_{k,k}(X) := [\Theta_{k,k}(x_i, x_j)]_{i,j \in [N]}$ corresponding to an output neuron $f_k, k \in [C]$ and every kernel $\Theta_{p,p}^{h}(X)$ corresponding to a last-layer neuron $h_p, p \in [n]$ is aligned with $\mathbf{Y}^{\top}\mathbf{Y}$ (see Figure 1, panes a-b). Additionally, the "non-diagonal" kernels $\Theta_{k,s}(X)$ and $\Theta_{k,s}^{h}(X), k \neq s$ are equal to zero (see Figure 1, panes c-d).[3] Moreover, if $\gamma_c \gg \gamma_n$ and $\kappa_c \gg \kappa_n$, Assumption 3.2 can be interpreted as *local elasticity* of DNNs, defined below.

**Definition 3.3** (Local elasticity [23]). *A classifier is said to be locally elastic (LE) if its prediction or feature representation on point $x_i^c$ from class $c \in [C]$ is not significantly affected by performing SGD updates on data points from classes $c' \neq c$.*

To see the relation between Assumption 3.2 and this definition, consider a Gradient Descent (GD) step of the output neuron $f_k, k \in [C]$ with step size $\eta$ performed on a single input $x_j^{c'}$ from class $c' \neq c$. By the chain rule, block-structured $\Theta$ implies locally-elastic predictions since

$$f^{t+1}(x_i^c) - f^t(x_i^c) = -\eta \Theta(x_i^c, x_j^{c'}) \frac{\partial \mathcal{L}(x_j^{c'})}{\partial f(x_j^{c'})} + O(\eta^2), \tag{6}$$

i.e., the magnitude of the GD step of $f(x_i^c)$ is determined by the value of $\Theta(x_i^c, x_j^{c'})$. Similarly, block-structured kernel $\Theta^h$ implies locally-elastic penultimate layer features because

$$h^{t+1}(x_i^c) - h^t(x_i^c) = -\eta \Theta^h(x_i^c, x_j^{c'}) \mathbf{W}^{\top} \frac{\partial \mathcal{L}(x_j^{c'})}{\partial f(x_j^{c'})} + O(\eta^2). \tag{7}$$

This observation provides a connection between our work and recent contributions suggesting a connection between NC and local elasticity [33, 53].

---

[2]We provide figures illustrating the NTK block structure on other architectures and datasets in Appendix C.

[3]We discuss possible relaxations to our main assumption, where the "non-diagonal" components of the last-layer kernel $\Theta_{k,s}^{h}$ are allowed to be non-zero, in Appendix D.

| Eigenvalue | Eigenvector | Multiplicity |
|---|---|---|
| $\lambda_{\text{single}} = \gamma_d - \gamma_c$ | $\mathbf{v}_i^c = \frac{1}{m-1}\big(\underbrace{m-1, -\mathbf{1}_{m-1}^\top}_{\text{index } i>0,\ \text{class } c<0}, \underbrace{\mathbf{0}_{N-m}^\top}_{\text{others} =0}\big)^\top$ | $N - C$ |
| $\lambda_{\text{class}} = \lambda_{\text{single}} + m(\gamma_c - \gamma_n)$ | $\mathbf{v}_c = \frac{1}{C-1}\big(\underbrace{(C-1)\mathbf{1}_m^\top}_{\text{class } c>0}, \underbrace{-\mathbf{1}_{N-m}^\top}_{\text{others} <0}\big)^\top$ | $C - 1$ |
| $\lambda_{\text{global}} = \lambda_{\text{class}} + N\gamma_n$ | $\mathbf{v}_0 = \mathbf{1}_N$ | $1$ |

Table 1: Eigendecomposition of the block-structured NTK.

## 4 Dynamics of DNNs with NTK Alignment

### 4.1 Convergence

As a warm up for our main results, we analyze the effects of the NTK block structure on the convergence of DNNs. Consider a GD update of an output neuron $f_k, k \in [C]$ with the step size $\eta$:

$$f_k^{t+1}(X) = f_k^t(X) - \eta\Theta_{k,k}(X)(f_k^t(X) - \mathbf{Y}_k) + O(\eta^2), \quad k = 1, \dots, C. \tag{8}$$

Note that we have taken into account that $\Theta_{k,s}$ is zero for $k \neq s$ by our assumption. Denote the residuals corresponding to $f_k$ as $\mathbf{r}_k^\top := f_k^\top(X) - \mathbf{Y}_k \in \mathbb{R}^N$. Then we have the following dynamics for the residuals vector:

$$\mathbf{r}_k^{t+1} = (1 - \eta\Theta_{k,k}(X))\mathbf{r}_k^t + O(\eta^2). \tag{9}$$

The eigendecomposition of the block-structured kernel $\Theta_{k,k}(X)$ provides important insights into this dynamics and is summarized in Table 1. We notice that the NTK has three distinct eigenvalues $\lambda_{\text{global}} \geq \lambda_{\text{class}} \geq \lambda_{\text{single}}$, which imply different convergence rates for certain components of the error. Moreover, the eigenvectors associated with each of these eigenvalues reveal the meaning of the error components corresponding to each convergence rate. Indeed, consider the projected dynamics with respect to eigenvector $\mathbf{v}_0$ and eigenvalue $\lambda_{\text{global}}$ from Table 1:

$$\langle \mathbf{r}_k^{t+1}, \mathbf{v}_0 \rangle = (1 - \eta\lambda_{\text{global}})\langle \mathbf{r}_k^t, \mathbf{v}_0 \rangle, \tag{10}$$

where we omitted $O(\eta^2)$ for clarity. Now notice that the projection of $\mathbf{r}_k^t$ onto the vector $\mathbf{v}_0$ is in fact proportional to the average residual over the training set:

$$\langle \mathbf{r}_k^t, \mathbf{v}_0 \rangle = \langle \mathbf{r}_k^t, \mathbf{1}_N \rangle = N\langle \mathbf{r}_k^t \rangle \tag{11}$$

where $\langle \cdot \rangle$ denotes the average over all the training samples $x_i \in X$. By a similar calculation, for all $c \in [C]$ and $i \in [m]$ we get interpretations of the remaining projections of the residual:

$$\langle \mathbf{r}_k^t, \mathbf{v}_c \rangle = \frac{N}{C-1}(\langle \mathbf{r}_k^t \rangle_c - \langle \mathbf{r}_k^t \rangle), \quad \langle \mathbf{r}_k^t, \mathbf{v}_i^c \rangle = \frac{m}{m-1}(\mathbf{r}_k^t(x_i^c) - \langle \mathbf{r}_k^t \rangle_c), \tag{12}$$

We where $\langle \cdot \rangle_c$ denotes the average over samples $x_i^c$ from class $c$, and $\mathbf{r}_k^\top(x_i^c)$ is the $k$th component of $f^\top(x_i^c) - y_i^c$. Combining (10), (11) and (12), we have the following convergence rates:

$$\langle \mathbf{r}_k^{t+1} \rangle = (1 - \eta\lambda_{\text{global}})\langle \mathbf{r}_k^t \rangle, \tag{13}$$

$$\langle \mathbf{r}_k^{t+1} \rangle_c - \langle \mathbf{r}_k^{t+1} \rangle = (1 - \eta\lambda_{\text{class}})(\langle \mathbf{r}_k^t \rangle_c - \langle \mathbf{r}_k^t \rangle), \tag{14}$$

$$\mathbf{r}_k^{t+1}(x_i^c) - \langle \mathbf{r}_k^{t+1} \rangle_c = (1 - \eta\lambda_{\text{single}})(\mathbf{r}_k^t(x_i^c) - \langle \mathbf{r}_k^t \rangle_c). \tag{15}$$

Overall, this means that the global mean $\langle \mathbf{r} \rangle$ of the residual converges first, then the class means, and finally the residual of each sample $\mathbf{r}(x_i^c)$. To simplify the notation, we define the following quantities:

$$\mathbf{R} = f(X) - \mathbf{Y} = [\mathbf{r}(x_1), \dots, \mathbf{r}(x_N)], \tag{16}$$

$$\mathbf{R}_{\text{class}} = \frac{1}{m}\mathbf{R}\mathbf{Y}^\top\mathbf{Y} = \underbrace{[\langle \mathbf{r} \rangle_1, \dots, \langle \mathbf{r} \rangle_C]}_{:=\mathbf{R}_1} \otimes \mathbf{1}_m^\top, \tag{17}$$

$$\mathbf{R}_{\text{global}} = \frac{1}{N}\mathbf{R}\mathbf{1}_N\mathbf{1}_N^\top = \langle \mathbf{r} \rangle \otimes \mathbf{1}_N^\top, \tag{18}$$

where $\mathbf{R} \in \mathbb{R}^{C \times N}$ is the matrix of residuals, $\mathbf{R}_{\text{class}} \in \mathbb{R}^{C \times N}$ are the residuals averaged over each class and stacked $m$ times, and $\mathbf{R}_{\text{global}} \in \mathbb{R}^{C \times N}$ are the residuals averaged over the whole training set stacked $N$ times. According to the previous discussion, $\mathbf{R}_{\text{global}}$ converges to zero at the fastest rate, while $\mathbf{R}$ converges at the slowest rate. The last phase, which we call the *end of training*, is when $\mathbf{R}_{\text{class}}$ and $\mathbf{R}_{\text{global}}$ have nearly vanished and can be treated as zero for the remaining training time. We will use this notion in several remarks, as well as in the proof of Theorem 5.2.

### 4.2 Gradient Flow Dynamics with Block-Structured NTK

We derive the dynamics of $\mathbf{H}, \mathbf{W}, \mathbf{b}$ under Assumption 3.2 in Theorem 4.1. One can see that the block-structured kernel greatly simplifies the complicated dynamics of DNNs and highlights the role of each of the residual components identified in Section 4.1. We consider gradient flow, which is close to gradient descent for sufficiently small step size [16], to reduce the complications caused by higher order terms. The proof is given in Appendix B.1.

**Theorem 4.1.** *Suppose Assumption 3.2 holds. Then the gradient flow dynamics of a DNN can be written as*

$$\begin{cases} \dot{\mathbf{H}} = & -\mathbf{W}^\top [(\kappa_d - \kappa_c)\mathbf{R} + (\kappa_c - \kappa_n)m\mathbf{R}_{\text{class}} + \kappa_n N \mathbf{R}_{\text{global}}] \\ \dot{\mathbf{W}} = & -\mathbf{R}\mathbf{H}^\top \\ \dot{\mathbf{b}} = & -\mathbf{R}_{\text{global}}\mathbf{1}_N. \end{cases} \tag{19}$$

We note that at the end of training, where $\mathbf{R}_{\text{class}}$ and $\mathbf{R}_{\text{global}}$ are zero, the system (19) reduces to

$$\dot{\mathbf{H}} = -(\kappa_d - \kappa_c)\nabla_{\mathbf{H}}\tilde{\mathcal{L}}, \qquad \dot{\mathbf{W}} = -\nabla_{\mathbf{W}}\tilde{\mathcal{L}}, \qquad \tilde{\mathcal{L}}(\mathbf{W}, \mathbf{H}) := \frac{1}{2}\|\mathbf{W}\mathbf{H} + \mathbf{b}\mathbf{1}_N^\top - \mathbf{Y}\|_F^2, \tag{20}$$

and $\dot{\mathbf{b}} = 0$. This system differs from the unconstrained features dynamics only by a factor of $\kappa_d - \kappa_c$ before $\mathbf{H}$. Moreover, such a form of the loss function also appears in the literature of implicit regularization [4, 6, 11], where the authors show that $\mathbf{W}\mathbf{H}$ converges to a low rank matrix.

## 5 NTK Alignment Drives Neural Collapse

The main goal of this work is to demonstrate how NC results from the NTK block structure. To this end, in Section 5.1 we further analyze the dynamics presented in Theorem 4.1, in Section 5.2 we derive the invariant of this training dynamics, and in Section 5.3 we finally derive NC.

### 5.1 Features Decomposition

We first decompose the features dynamics presented in Theorem 4.1 into two parts: $\mathbf{H}_1$, which lies in the subspace of the labels $\mathbf{Y}$, and $\mathbf{H}_2$, which is orthogonal to the labels and eventually vanishes. To achieve this, note that the SVD of $\mathbf{Y}$ has the following form:

$$\mathbf{P}^\top \mathbf{Y}\mathbf{Q} = \begin{bmatrix} \sqrt{m}\mathbb{I}_C, \mathbb{O} \end{bmatrix}, \tag{21}$$

where $\mathbb{O} \in \mathbb{R}^{C \times (N-C)}$ is a matrix of zeros, and $\mathbf{P} \in \mathbb{R}^{C \times C}$ and $\mathbf{Q} \in \mathbb{R}^{N \times N}$ are orthogonal matrices. Moreover, we can choose $\mathbf{P}$ and $\mathbf{Q}$ such that $\mathbf{P} = \mathbb{I}_C$ and

$$\mathbf{Q} = \begin{bmatrix} \mathbf{Q}_1, \mathbf{Q}_2 \end{bmatrix}, \quad \mathbf{Q}_1 = \frac{1}{\sqrt{m}}\mathbb{I}_C \otimes \mathbf{1}_m \in \mathbb{R}^{N \times C}, \quad \mathbf{Q}_2 = \mathbb{I}_C \otimes \tilde{\mathbf{Q}}_2 \in \mathbb{R}^{N \times (N-C)}, \tag{22}$$

where $\otimes$ is the Kronecker product. Note that by orthogonality, $\tilde{\mathbf{Q}}_2 \in \mathbb{R}^{m \times (m-1)}$ has full rank and $\mathbf{1}_m^\top \tilde{\mathbf{Q}}_2 = \mathbb{O}$. We can now decompose $\mathbf{H}\mathbf{Q}$ into two components as follows:

$$\mathbf{H}\mathbf{Q} = \sqrt{m}[\mathbf{H}_1, \mathbf{H}_2], \quad \mathbf{H}_1 = \frac{1}{\sqrt{m}}\mathbf{H}\mathbf{Q}_1, \quad \mathbf{H}_2 = \frac{1}{\sqrt{m}}\mathbf{H}\mathbf{Q}_2. \tag{23}$$

The following equations reveal the meaning of these two components:

$$\mathbf{H}_1 = \begin{bmatrix} \langle h \rangle_1, \dots, \langle h \rangle_C \end{bmatrix}, \quad \mathbf{H}_2 = \frac{1}{\sqrt{m}} \begin{bmatrix} \mathbf{H}^{(1)}\tilde{\mathbf{Q}}_2, \dots, \mathbf{H}^{(C)}\tilde{\mathbf{Q}}_2 \end{bmatrix}, \tag{24}$$

where $\langle h \rangle_c \in \mathbb{R}^n$ is the mean of $h$ over inputs $x_i^c$ from class $c \in [C]$, and $\mathbf{H}^{(c)} \in \mathbb{R}^{n \times m}$ is the submatrix of $\mathbf{H}$ corresponding to samples of class $c$, i.e., $\mathbf{H} = \begin{bmatrix} \mathbf{H}^{(1)}, \dots, \mathbf{H}^{(C)} \end{bmatrix}$. We see that $\mathbf{H}_1$

is simply the matrix of the last-layer features' class means, which is prominent in the NC literature. We also see that the columns of $\mathbf{H}^{(c)}\tilde{\mathbf{Q}}_2$ are $m-1$ different linear combinations of $m$ vectors $h(x_i^c)$, $i \in [m]$. Moreover, the coefficients of each of these linear combinations sum to zero by the choice of $\tilde{\mathbf{Q}}_2$. Therefore, $\mathbf{H}_2$ must reduce to zero in case of variability collapse (NC1), when all the feature vectors within the same class become equal. We prove that $\mathbf{H}_2$ indeed vanishes in DNNs with block-structured NTK as part of our main result (Theorem 5.2).

## 5.2 Invariant

We now use the former decomposition of the last-layer features to further simplify the dynamics and deduce a training invariant in Theorem 5.1. The proof is given in Appendix B.2.

**Theorem 5.1.** *Suppose Assumption 3.2 holds. Define $\mathbf{H}_1$ and $\mathbf{H}_2$ as in* (23). *Then the class-means of the residuals (defined in* (17)*) are given by $\mathbf{R}_1 = \mathbf{W}\mathbf{H}_1 + \mathbf{b}\mathbf{1}_C^\top - \mathbb{I}_C$, and the training dynamics of the DNN can be written as*

$$\begin{cases} \dot{\mathbf{H}}_1 &= -\mathbf{W}^\top\mathbf{R}_1(\mu_{\text{class}}\mathbb{I}_C + \kappa_n m \mathbf{1}_C\mathbf{1}_C^\top) \\ \dot{\mathbf{H}}_2 &= -\mu_{\text{single}}\mathbf{W}^\top\mathbf{W}\mathbf{H}_2 \\ \dot{\mathbf{W}} &= -m(\mathbf{R}_1\mathbf{H}_1^\top + \mathbf{W}\mathbf{H}_2\mathbf{H}_2^\top) \\ \dot{\mathbf{b}} &= -m\mathbf{R}_1\mathbf{1}_C, \end{cases} \tag{25}$$

*where $\mu_{\text{single}} := \kappa_d - \kappa_c$ and $\mu_{\text{class}} := \mu_{\text{single}} + m(\kappa_c - \kappa_n)$ are the two smallest eigenvalues of the kernel $\Theta_{k,k}^h(X)$ for any $k \in [n]$. Moreover, the quantity*

$$\mathbf{E} := \frac{1}{m}\mathbf{W}^\top\mathbf{W} - \frac{1}{\mu_{\text{class}}}\mathbf{H}_1(\mathbb{I}_C - \alpha\mathbf{1}_C\mathbf{1}_C^\top)\mathbf{H}_1^\top - \frac{1}{\mu_{\text{single}}}\mathbf{H}_2\mathbf{H}_2^\top \tag{26}$$

*is invariant in time. Here $\alpha := \frac{\kappa_n m}{\mu_{\text{class}} + C\kappa_n m}$.*

We note that the invariant $\mathbf{E}$ derived here resembles the conservation laws of *hyperbolic* dynamics that take the form $\mathbf{E}_{\text{hyp}} := a^2 - b^2 = const$ for time-dependent quantities $a$ and $b$. Such dynamics arise when gradient flow is applied to a loss function of the form $\mathcal{L}(a, b) := (ab - q)^2$ for some $q$. Since the solutions of such minimization problems, given by $ab = q$, exhibit symmetry under scaling $a \to \gamma a, b \to b/\gamma$, the value of the invariant $\mathbf{E}_{\text{hyp}}$ uniquely specifies the hyperbola followed by the solution. In machine learning theory, hyperbolic dynamics arise as the gradient flow dynamics of linear DNNs [42], or in matrix factorization problems [3, 15]. Moreover, the end of training dynamics defined in (20) has a hyperbolic invariant given by

$$\mathbf{E}_{\text{eot}} := \mathbf{W}^\top\mathbf{W} - \frac{1}{\mu_{\text{single}}}\mathbf{H}\mathbf{H}^\top. \tag{27}$$

Therefore, the final phase of training exhibits a typical behavior for the hyperbolic dynamics, which is also characteristic for the unconstrained features models [21, 38]. Namely, "scaling" $\mathbf{W}$ and $\mathbf{H}$ by an invertible matrix does not affect the loss value but changes the dynamic's invariant. On the other hand, minimizing the invariant $\mathbf{E}_{\text{eot}}$ has the same effect as joint regularization of $\mathbf{W}$ and $\mathbf{H}$ [48].

However, we also note that our invariant $\mathbf{E}$ provides a new, more comprehensive look at the DNNs' dynamics. While unconstrained features models effectively make assumptions on the end-of-training invariant $\mathbf{E}_{\text{eot}}$ to derive NC [21, 38, 48], our dynamics control the value of $\mathbf{E}_{\text{eot}}$ through the more general invariant $\mathbf{E}$. This way we connect the properties of end-of-training hyperbolic dynamics with the previous stages of training.

## 5.3 Neural Collapse

We are finally ready to state and prove our main result in Theorem 5.2 about the emergence of NC in DNNs with NTK alignment. We include the proof in Appendix B.3.

**Theorem 5.2.** *Assume that the NTK has a block structure as defined in Assumption 3.2. Then the DNN's training dynamics are given by the system of equations in* (25)*. Assume further that the last-layer features are centralized, i.e $\langle h \rangle = 0$, and the dynamics invariant* (26) *is zero, i.e., $\mathbf{E} = \mathbb{O}$. Then the DNN's dynamics exhibit neural collapse as defined in (NC1)-(NC4).*

Below we provide several important remarks and discuss the implications of this result:

**(1) Zero invariant assumption:** We assume that the invariant (26) is zero in Theorem 5.2 for simplicity and consistency with the literature. Indeed, similar assumptions arise in matrix decomposition papers, where zero invariant guarantees "balance" of the problem [3, 15]. However, our proofs in fact only require a weaker assumption that the invariant terms containing features $\mathbf{H}$ are aligned with the weights $\mathbf{W}^\top \mathbf{W}$, i.e.

$$\mathbf{W}^\top \mathbf{W} \propto \frac{1}{\mu_{\text{class}}} \mathbf{H}_1 \mathbf{H}_1^\top - \frac{1}{\mu_{\text{single}}} \mathbf{H}_2 \mathbf{H}_2^\top, \tag{28}$$

where we have taken into account our assumption on the zero global mean $\langle h \rangle = 0$.

**(2) Necessity of the invariant assumption:** The relaxed assumption on the invariant (28) is necessary for the emergence of NC in DNNs with block-structured NTK. Indeed, NC1 implies $\mathbf{H}_2 = \mathbb{O}$, and NC3 implies $\mathbf{H}_1 \mathbf{H}_1^\top \propto \mathbf{W}^\top \mathbf{W}$. Therefore, DNNs that do not satisfy this assumption do not display NC. Our numerical experiments described in Section 6 strongly support this insight (see Figure 2, panes a-e). Thus, we believe that the invariant derived in this work characterizes the difference between models that do and do not exhibit NC.

**(3) Zero global mean assumption:** We note that the zero global mean assumption $\langle h \rangle = 0$ in Theorem 5.2 ensures that the biases are equal to $\mathbf{b} = \frac{1}{C} \mathbf{1}_C$ at the end of training. This assumption is common in the NC literature [21, 38] and is well-supported by our numerical experiments (see figures in Appendix C, pane i). Indeed, modern DNNs typically include certain normalization (e.g. through batch normalization layers) to improve numerical stability, and closeness of the global mean to zero is a by-product of such normalization.

**(4) General biases case:** Discarding the zero global mean assumption allows the biases $\mathbf{b}$ to take an arbitrary form. In this general case, the following holds for the matrix of weights:

$$(\mathbf{W}\mathbf{W}^\top)^2 = \frac{m}{\mu_{\text{class}}} \Big( \mathbb{I}_C - \alpha \mathbf{1}_C \mathbf{1}_C^\top + (1 - \alpha C)(C \mathbf{b}\mathbf{b}^\top - \mathbf{b} \mathbf{1}_C^\top - \mathbf{1}_C \mathbf{b}^\top) \Big). \tag{29}$$

For optimal biases $\mathbf{b} = \frac{1}{C} \mathbf{1}_C$, this reduces to the ETF structure that emerges in NC. Moreover, if biases are all equal, i.e. $\mathbf{b} = \beta \mathbf{1}_C$ for some $\beta \in \mathbb{R}$, the centralized class means still form an ETF (i.e., NC2 holds), and the weights exhibit a certain symmetric structure given by

$$\mathbf{W}\mathbf{W}^\top \propto \Big( \mathbb{I}_C - \gamma \mathbf{1}_C \mathbf{1}_C^\top \Big), \quad \mathbf{M}^\top \mathbf{M} \propto \Big( \mathbb{I}_C - \frac{1}{C} \mathbf{1}_C \mathbf{1}_C^\top \Big), \tag{30}$$

where $\gamma := \frac{1}{C}(1 - |1 - \beta C|\sqrt{1 - \alpha C}) < \frac{1}{C}$. The proof and a discussion of this result are given in Appendix B.4. In general, the angles of these two frames are different, and thus NC3 does not hold. This insight leads us to believe that normalization is an important factor in the emergence of NC.

**(5) Partial NC:** Our proofs and the discussion suggest that all the four phenomena that form NC do not have to always coincide. In particular, our proof of NC1 only requires the block-structured NTK and the invariant to be P.S.D, which is much weaker than the total set of assumptions in Theorem 5.2. Therefore, variability collapse can occur in models that do not exhibit the ETF structure of the class-means or the duality of the weights and the class means. Moreover, as shown above, NC2 can occur when NC3 does not, i.e., the ETF structure of the class means does not imply duality.

## 6 Experiments

We conducted large-scale numerical experiments to support our theory. While we only showcase our results on a single dataset-architecture pair in the main text (see Figure 2) and refer the rest to the appendix, the following discussion covers all our experiments.

**Datasets and models.** Following the seminal NC paper [39], we use three canonical DNN architectures: VGG [46], ResNet [24] and DenseNet [26]. Our datasets are MNIST [35], FashionMNIST [51] and CIFAR10 [34]. We choose VGG11 for MNIST and FashionMNIST, and VGG16 for CIFAR10. We add batch normalization after every layer in the VGG architecture, set dropout to zero and choose the dimensions of the two fully-connected layers on the top of the network as $512$ and $256$. We use ResNet20 architecture described in the original ResNet paper [24], and DenseNet40 with bottleneck layers, growth $k = 12$, and zero dropout for all the datasets.

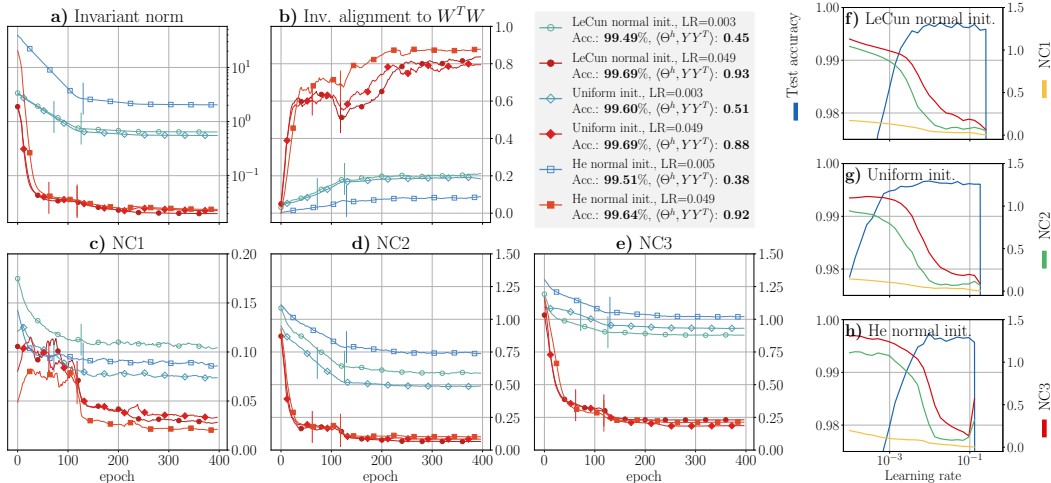

Figure 2: ResNet20 trained on MNIST with three initialization settings and varying learning rates (see Section 6 for details). We chose a model that exhibits NC (red lines, filled markers) and a model that does not exhibit NC (blue lines, empty markers) for each initialization. The vertical lines indicate the epoch when the training accuracy reaches 99.9% (over the last 10 batches). **a)** Frobenious norm of the invariant $\|\mathbf{E}\|_F$. **b)** Alignment of the invariant terms as defined in (28). **c)** NC1: standard deviation of $h(x_i^c)$ averaged over classes. **d)** NC2: $\|\mathbf{M}^\top\mathbf{M}/\|\mathbf{M}^\top\mathbf{M}\|_F - \Phi\|_F$, where $\Phi$ is an ETF. **e)** NC3: $\|\mathbf{W}^\top/\|\mathbf{W}\|_F - \mathbf{M}/\|\mathbf{M}\|_F\|_F$. The legend displays the test accuracy achieved by each model and the last-layer features kernel alignment given by $\langle \Theta^h/\|\Theta^h\|_F, \mathbf{Y}^\top\mathbf{Y}/\|\mathbf{Y}^\top\mathbf{Y}\|_F\rangle_F$. The curves in panes a-e are smoothed by Savitzky–Golay filter with polynomial degree 1 over window of size 10. Panes **f**, **g** and **h** show the NC metrics and the test accuracy as functions of the learning rate.

**Optimization and initialization.** We use SGD with Nesterov momentum $0.9$ and weight decay $5 \times 10^{-4}$. Every model is trained for 400 epochs with batches of size 120. To be consistent with the theory, we balance the batches exactly. We train every model with a set of initial learning rates spaced logarithmically in the range $\eta \in [10^{-4}, 10^{0.25}]$. The learning rate is divided by 10 every 120 epochs. On top of the varying learning rates, we try three different initialization settings for every model: **(a)** LeCun normal initialization (default in Flax), **(b)** uniform initialization on $[-\sqrt{k}, \sqrt{k}]$, where $k = 1/n_{\ell-1}$ for a linear layer, and $k = 1/(Kn_{\ell-1})$ for a convolutional layer, where $K$ is the convolutional kernel size (default in PyTorch), **(c)** He normal initialization in `fan_out` mode.

**Results.** Our experiments confirm the validity of our assumptions and the emergence of NC as their result. Specifically, we make the following observations:

- While most of the DNNs that achieve high test performance exhibit NC, we are able to identify DNNs with comparable performance that do not exhibit NC (see Figure 2, panes f-h). We note that such models still achieve near-zero error on the training set in our setup.

- Comparing DNNs that do and do not exhibit NC, we find that our assumption on the invariant (see Theorem 5.2 and (28)) holds only for the models with NC (see Figure 2, panes a-e). This confirms our reasoning about the necessity of the invariant assumption for NC emergence.

- The kernels $\Theta$ and $\Theta^h$ are strongly aligned with the labels $\mathbf{Y}^\top\mathbf{Y}$ in the models with the best performance, which is in agreement with the NTK alignment literature and justifies our assumption on the NTK block structure.

We include the full range of experiments along with the implementation details and the discussion of required computational resources in Appendix C. Specifically, we present a figure analogous to Figure 2 for every considered dataset-architecture pair. Additionally, we report the norms of matrices $\mathbf{H}_1\mathbf{H}_1^\top$, $\mathbf{H}_2\mathbf{H}_2^\top$, and $\langle h\rangle\langle h\rangle^\top$, as well as the alignment of both the NTK $\Theta$ and the last-layer features kernel $\Theta^h$ in the end of training, to further justify our assumptions.

# 7 Conclusions and Broad Impact

This work establishes the connection between NTK alignment and NC, and thus provides a mechanistic explanation for the emergence of NC within realistic DNNs' training dynamics. It also contributes to the underexplored line of research connecting NC and local elasticity of DNNs' training dynamics.

The primary implication of this research is that it exposes the potential to study NC through the lens of NTK alignment. Indeed, previous works on NC focus on the top-down approach (layer-peeled models) [18, 21, 38, 48], and fundamentally cannot explain how NC develops through earlier layers of a DNN and what are the effects of depth. On the other hand, NTK alignment literature focuses on the alignment of individual layers [7], and recent theoretical results even quantify the role of each hidden layer in the final alignment [37]. Therefore, we believe that the connection between NTK alignment and NC established in this work provides a conceptually new method to study NC.

Moreover, this work introduces a novel approach to facilitate theoretical analysis of DNNs' training dynamics. While most theoretical works consider the NTK in the infinite-width limit to simplify the dynamics [1, 20, 28, 49], our analysis shows that making reasonable assumptions on the empirical NTK can also lead to tractable dynamics equations and new theoretical results. Thus, we believe that the analysis of DNNs' training dynamics based on the properties of the empirical NTK is a promising approach also beyond NC research.

# 8 Limitations and Future Work

The main limitation of this work is the simplifying Assumption 3.2 on the kernel structure. While the NTK of well-trained DNNs indeed has an approximate block structure (as we discuss in detail in Section 3), the NTK values also tend to display high variance in real DNNs [22, 44]. Thus, we believe that adding stochasticity to the dynamics considered in this paper is a promising direction for the future work. Moreover, the empirical NTK exhibits so-called specialization, i.e., the kernel matrix corresponding to a certain output neurons aligns more with the labels of the corresponding class [45]. In block-structured kernels, specialization implies different values in blocks corresponding to different classes. Thus, generalizing our theory to block-structured kernels with specialization is another promising short-term research goal. In addition, our theory relies on the assumption that the dataset (or the training batch) is balanced, i.e., all the classes have the same number of samples. Accounting for the effects of non-balanced datasets within the dynamics of DNNs with block-structured NTK is another possible future work direction.

More generally, we believe that empirical observations are essential to demistify the DNNs' training dynamics, and there are still many unknown and interesting connections between seemingly unrelated empirical phenomena. Establishing new theoretical connections between such phenomena is an important objective, since it provides a more coherent picture of the deep learning theory as a whole.

## Acknowledgments and Disclosure of Funding

R. Giryes and G. Kutyniok acknowledge support from the LMU-TAU - International Key Cooperation Tel Aviv University 2023. R. Giryes is also grateful for partial support by ERC-StG SPADE grant no. 757497. G. Kutyniok is grateful for partial support by the Konrad Zuse School of Excellence in Reliable AI (DAAD), the Munich Center for Machine Learning (BMBF) as well as the German Research Foundation under Grants DFG-SPP-2298, KU 1446/31-1 and KU 1446/32-1 and under Grant DFG-SFB/TR 109, Project C09 and the Federal Ministry of Education and Research under Grant MaGriDo.

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

# A    Related works

**NC with MSE loss.**    NC was first introduced for DNNs with cross-entropy (CE) loss, which is commonly used in classification problems [39]. Since then, numerous papers discussed NC with MSE loss, which provides more opportunities for theoretical analysis, especially after the MSE loss was shown to perform on par with CE loss for classification tasks [14, 29].

Most previous works on MSE-NC adopt the so-called unconstrained features model [21, 38, 48]. In this model, the last-layer features $\mathbf{H}$ are free variables that are directly optimized during training, i.e., the features do not depend on the input data or the DNN's trainable parameters. Fang *et al.* [18] also introduced a generalization of this approach called $N$-layer-peeled model, where features of the $N$-th-to-last layer are free variables, and studied the 1-layer-peeled model (equivalent to the unconstrained features model) with CE loss as a special case.

One line of research on MSE-NC in unconstrained/layer-peeled models aims to derive global minimizers of optimization problems associated with DNNs [17, 18, 48]. In particular, Tirer *et al.* [48] showed that global minimizers of the MSE loss with regularization of both $\mathbf{H}$ and $\mathbf{W}$ exhibit NC. Moreover, Ergen & Pilanci [17] showed that NC emerges in global minimizers of optimization problems with general convex loss in the context of the 2-layer-peeled model. In comparison to our work, these contributions do not consider the training dynamics of DNNs, i.e., they do not discuss whether and how the model converges to the optimal solution.

Another line of research on MSE-NC explicitly considers the dynamics of the unconstrained features models [21, 38]. In particular, Han *et al.* [21] considered the gradient flow of the unconstrained renormalized features along the "central path", where the classifier is assumed to take the form of the optimal least squares (OLS) solution for given features $\mathbf{H}$. Under this assumption, they derive a closed-form dynamics that implies NC. While they empirically show that DNNs are close to the central path in certain scenarios, they do not provide a theoretical justification for this assumption. The dynamics considered in their work is also distinct from the standard gradient flow dynamics of DNNs considered in our work. On the other hand, an earlier work by Mixon *et al.* [38] considered the gradient flow dynamics of the unconstrained features model, which is equivalent (up to rescaling) to the end-of-training dynamics (20) that we discuss in Sections 4.2 and 5.2. Their work relies on the linearization of these dynamics to derive a certain subspace, which appears to be an invariant subspace of the non-linearized unconstrained features model dynamics. Then they show that minimizers of the loss from this subspace exhibit NC. We note that, in terms of our paper, assuming that the unconstrained features model dynamics follow a certain invariant subspace means making assumptions on the end-of-training invariant (27). In comparison to these works, we make a step towards realistic DNNs dynamics by considering the standard gradient flow of DNNs simplified by Assumption 3.2 on the NTK structure, which is supported by the extensive research on NTK alignment [7, 9, 44, 45]. In our setting, the NTK captures the dependence of the features on the training data, which is missing in the unconstrained features model. Moreover, while other works focus only on the dynamics that converge to NC, we show that DNNs with MSE loss may not exhibit NC in certain settings, and the invariant of the dynamics (26) characterizes the difference between models that do and do not converge to NC.

Notably, works by Poggio & Liao [41] adopt a model different from the unconstrained features model to analyze gradient flow of DNNs. They consider the dynamics of homogeneous DNNs, in particular ReLU networks without biases, with normalization of the weights matrices and weights regularization. The goal of weights normalization in their model is to imitate the effects of batch normalization in DNNs training. In this model, certain fixed points of the gradient flow exhibit NC. While the approach taken in their work captures the dependence of the features on the data and the DNN's parameters, it fundamentally relies on the homogeneity of the DNN's output function. However, most DNNs that exhibit NC in practice are not homogeneous due to biases and skip-connections.

**NC and local elasticity.**    A recent extensive survey of NC literature [33] discussed local elasticity as a possible mechanism behind the emergence of NC, which has not been sufficiently explored up until now. One of the few works in this research direction is by Zhang *et al.* [53], who analyzed the so-called locally-elastic stochastic differential equations (SDEs) and showed the emergence of NC in their solutions. They model local elasticity of the dynamics through an effect matrix, which has only two distinct values: a larger intra-class value and a smaller inter-class value. These values characterize how much influence samples from one class have on samples from other classes in the SDEs. While the aim of their work is to imitate DNNs' training dynamics through SDEs, the authors

do not provide any explicit connection between their dynamics and real gradient flow dynamics of DNNs. On the other hand, we derive our dynamics directly from the gradient flow equations and connect local elasticity to the NTK, which is a well-studied object in the deep learning theory.

Another work by Tirer *et al.* [50] provided a perturbation analysis of NC to study "inexact collapse". They considered a minimization problem with MSE loss, regularization of $\mathbf{H}$ and $\mathbf{W}$, and additional regularization of the distance between $\mathbf{H}$ and a given matrix of initial features. In the "near-collapse" setting, i.e., when the initial features are already close to collapse, they showed that the optimal features can be obtained from the initial features by a certain linear transformation with a block structure, where the intra-class effects are stronger than the inter-class ones. While this transformation matrix resembles the block-structured effect matrices in locally-elastic training dynamics, it does not originate from the gradient flow dynamics of DNNs and is not related to the NTK.

## B  Proofs

### B.1  Proof of Theorem 4.1

*Proof of Theorem 4.1.* We will first derive the dynamics of $h_s(x_i^c)$, which is the $s$-th component of the last-layer features vector on sample $x_i^c \in X$ from class $c \in [C]$. Let $\mathbf{w} \in \mathbb{R}^P$ be the trainable parameters of the network stretched into a single vector. Then its gradient flow dynamics is given by

$$\dot{\mathbf{w}} = -\nabla_{\mathbf{w}}\mathcal{L}(f) = -\sum_{k=1}^{C}\sum_{i'=1}^{N}(f(X)_{ki'} - \mathbf{Y}_{ki'})\nabla_{\mathbf{w}}f(X)_{ki'}, \tag{31}$$

where $\nabla_{\mathbf{w}}f(X)_{ki'} \in \mathbb{R}^P$ is the component of the DNN's Jacobian corresponding to output neuron $k$ and the input sample $x_{i'}^{c'}$. Since entries of $f(X)$ can be written as

$$f(X)_{ki'} = \sum_{s'=1}^{n}\mathbf{W}_{ks'}\mathbf{H}_{s'i'} + \mathbf{b}_k = \sum_{s'=1}^{n}\mathbf{W}_{ks'}h_{s'}(x_{i'}^{c'}) + \mathbf{b}_k, \tag{32}$$

we obtain

$$\dot{\mathbf{w}} = -\sum_{k=1}^{C}\sum_{i'=1}^{N}\sum_{s'=1}^{n}(f(X)_{ki'} - \mathbf{Y}_{ki'})\nabla_{\mathbf{w}}(\mathbf{W}_{ks'}h_{s'}(x_{i'}^{c'}) + \mathbf{b}_k). \tag{33}$$

By chain rule, we have $\dot{h}_s(x_i^c) = \langle \nabla_{\mathbf{w}}h_s(x_i^c), \dot{\mathbf{w}} \rangle$. Then, taking into account that

$$\langle \nabla_{\mathbf{w}}h_s(x_i^c), \nabla_{\mathbf{w}}(\mathbf{W}_{ks'}h_{s'}(x_{i'}^{c'}) + \mathbf{b}_k) \rangle = \mathbf{W}_{ks'}\langle \nabla_{\mathbf{w}}h_s(x_i^c), \nabla_{\mathbf{w}}h_{s'}(x_{i'}^{c'}) \rangle, \tag{34}$$

and that $\langle \nabla_{\mathbf{w}}h_s(x_i^c), \nabla_{\mathbf{w}}h_{s'}(x_{i'}^{c'}) \rangle = \Theta_{s,s'}^h(x_i^c, x_{i'}^{c'})$ by definition of $\Theta^h$, we have

$$\dot{h}_s(x_i^c) = -\sum_{k=1}^{C}\sum_{i'=1}^{N}\sum_{s'=1}^{n}(f(X)_{ki'} - \mathbf{Y}_{ki'})\mathbf{W}_{ks'}\Theta_{s,s'}^h(x_i^c, x_{i'}^{c'}). \tag{35}$$

Now by Assumption 3.2 we have $\Theta_{s,s'}^h = 0$ if $s \neq s'$. Therefore, the above expression simplifies to

$$\dot{h}_s(x_i^c) = -\sum_{i'=1}^{N}\Theta_{s,s}^h(x_i^c, x_{i'}^{c'})\sum_{k=1}^{C}(f(X)_{ki'} - \mathbf{Y}_{ki'})\mathbf{W}_{ks}$$

$$= -\sum_{i'=1}^{N}[\mathbf{W}^\top(\mathbf{W}\mathbf{H} + \mathbf{b}\mathbf{1}_N^\top - \mathbf{Y})]_{si'}\Theta_{s,s}^h(x_i^c, x_{i'}^{c'}).$$

To express $\dot{\mathbf{H}} = [\dot{h}_s(x_i^c)]_{s,i} \in \mathbb{R}^{n \times N}$ in matrix form, it remains to express $\Theta_{s,s}^h(x_i^c, x_{i'}^{c'})$ as the $(i', i)$-th entry of some matrix. We will separate the sum into three cases: 1) $i = i'$, 2) $i \neq i'$ and $c = c'$, and 3) $c \neq c'$. According to Assumption 3.2, the first case corresponds to the multiple of identity $\kappa_d\mathbb{I}_N$. The second corresponds to the block matrix of size $m$ with zeros on the diagonal,

which can be written as $\kappa_c(\mathbf{Y}^\top\mathbf{Y} - \mathbb{I}_N)$. The third matrix equals to $\kappa_n(\mathbf{1}_N\mathbf{1}_N^\top - \mathbf{Y}^\top\mathbf{Y})$. Therefore we can express the dynamics of $\mathbf{H}$ as follows:

$$\dot{\mathbf{H}} = - [\mathbf{W}^\top(\mathbf{WH} + \mathbf{b}\mathbf{1}_N^\top - \mathbf{Y})][\kappa_d\mathbb{I} + \kappa_c(\mathbf{Y}^\top\mathbf{Y} - \mathbb{I}) + \kappa_n(\mathbf{1}_N\mathbf{1}_N^\top - \mathbf{Y}^\top\mathbf{Y})]$$
$$= - (\kappa_d - \kappa_c)\mathbf{W}^\top(\mathbf{WH} + \mathbf{b}\mathbf{1}_N^\top - \mathbf{Y})$$
$$- (\kappa_c - \kappa_n)\mathbf{W}^\top(\mathbf{WH}\mathbf{Y}^\top\mathbf{Y} + m\mathbf{b}\mathbf{1}_N^\top - m\mathbf{Y})$$
$$- \kappa_n\mathbf{W}^\top(\mathbf{WH}\mathbf{1}_N\mathbf{1}_N^\top + N\mathbf{b}\mathbf{1}_N^\top - \frac{N}{C}\mathbf{1}_C\mathbf{1}_N^\top).$$

Now we notice that $\mathbf{HY}^\top\mathbf{Y}/m$ is the matrix of stacked class means repeated $m$ times each and $\mathbf{H}\mathbf{1}_N\mathbf{1}_N^\top/N$ is a matrix of the global mean repeated $N$ times. Therefore, we have

$$\mathbf{WH}\mathbf{Y}^\top\mathbf{Y} + m\mathbf{b}\mathbf{1}_N^\top - m\mathbf{Y} = m\mathbf{R}_{\text{class}},$$
$$\mathbf{WH}\mathbf{1}_N\mathbf{1}_N^\top + N\mathbf{b}\mathbf{1}_N^\top - \frac{N}{C}\mathbf{1}_C\mathbf{1}_N^\top = N\mathbf{R}_{\text{global}}$$

according to the definitions of global and class-mean residuals in (18) and (17).

The expressions for the gradient flow dynamics of $\mathbf{W}$ and $\mathbf{b}$ follow directly from the derivatives of $f(X)$ w.r.t. $\mathbf{W}$ and $\mathbf{b}$. This completes the proof. $\qquad\square$

## B.2 Proof of Theorem 5.1

*Proof of Theorem 5.1.* Recall from (23) in Section 5.1 that we have the following decomposition

$$\mathbf{HQ} = \sqrt{m}[\mathbf{H}_1, \mathbf{H}_2], \quad \mathbf{H}_1 = \frac{1}{\sqrt{m}}\mathbf{HQ}_1, \quad \mathbf{H}_2 = \frac{1}{\sqrt{m}}\mathbf{HQ}_2$$

with orthogonal $\mathbf{Q} = [\mathbf{Q}_1, \mathbf{Q}_2] \in \mathbb{R}^{N\times N}$. We now artificially add $\mathbf{QQ}^\top(= \mathbb{I}_N)$ to the dynamics (19) in Theorem 4.1 and obtain

$$(36) \quad \begin{cases} \dot{\mathbf{H}}\mathbf{Q} = & -(\kappa_d - \kappa_c)\mathbf{W}^\top(\mathbf{WHQ} + \mathbf{b}\mathbf{1}_N^\top\mathbf{Q} - Y\mathbf{Q}) \\ & -(\kappa_c - \kappa_n)m\mathbf{W}^\top(\frac{1}{m}\mathbf{WHQ}\mathbf{Q}^\top\mathbf{Y}^\top\mathbf{YQ} + \mathbf{b}\mathbf{1}_N^\top\mathbf{Q} - Y\mathbf{Q}) \\ & -\kappa_n N\mathbf{W}^\top(\frac{1}{N}\mathbf{WHQ}\mathbf{Q}^\top\mathbf{1}_N\mathbf{1}_N^\top\mathbf{Q} + \mathbf{b}\mathbf{1}_N^\top\mathbf{Q} - \frac{1}{C}\mathbf{1}_C\mathbf{1}_N^\top\mathbf{Q}) \\ \dot{\mathbf{W}} = & -(\mathbf{WHQ} + \mathbf{b}\mathbf{1}_N^\top\mathbf{Q} - Y\mathbf{Q})\mathbf{Q}^\top\mathbf{H}^\top \\ \dot{\mathbf{b}} = & -(\mathbf{WHQ} + \mathbf{b}\mathbf{1}_N^\top\mathbf{Q} - Y\mathbf{Q})\mathbf{Q}^\top\mathbf{1}_N. \end{cases}$$

Let us simplify the expression. Since $\mathbf{Q}_1 = \frac{1}{\sqrt{m}}\mathbb{I}_C \otimes \mathbf{1}_m$ and $\mathbf{Q}_2 = \mathbb{I}_C \otimes \tilde{\mathbf{Q}}_2$, we have

$$\mathbf{1}_N^\top\mathbf{Q} = \sqrt{m}[\mathbf{1}_C^\top, \mathbb{O}], \quad \mathbf{YQ} = \sqrt{m}[\mathbb{I}_C, \mathbb{O}]. \tag{37}$$

Plugging (37) into (36), we see the dynamics can be decomposed into

$$(38) \quad \begin{cases} \dot{\mathbf{H}}_1 = & -(\kappa_d - \kappa_c)\mathbf{W}^\top(\mathbf{WH}_1 + \mathbf{b}\mathbf{1}_C^\top - \mathbb{I}_C) \\ & -(\kappa_c - \kappa_n)m\mathbf{W}^\top(\mathbf{WH}_1 + \mathbf{b}\mathbf{1}_C^\top - \mathbb{I}_C) \\ & -\kappa_n N\mathbf{W}^\top(\frac{1}{C}\mathbf{WH}_1\mathbf{1}_C\mathbf{1}_C^\top + \mathbf{b}\mathbf{1}_C^\top - \frac{1}{C}\mathbf{1}_C\mathbf{1}_C^\top) \\ \dot{\mathbf{H}}_2 = & -(\kappa_d - \kappa_c)\mathbf{W}^\top\mathbf{WH}_2 \\ \dot{\mathbf{W}} = & -m(\mathbf{WH}_1 + \mathbf{b}\mathbf{1}_C^\top - \mathbb{I}_C)\mathbf{H}_1^\top - m\mathbf{WH}_2\mathbf{H}_2^\top \\ \dot{\mathbf{b}} = & -m(\mathbf{WH}_1 + \mathbf{b}\mathbf{1}_C^\top - \mathbb{I}_C)\mathbf{1}_C. \end{cases}$$

To further simplify (38), we define the following quantities

$$\mu_{\text{single}} := \kappa_d - \kappa_c, \quad \mu_{\text{class}} := \mu_{\text{single}} + m(\kappa_c - \kappa_n), \quad \mathbf{R}_1 := \mathbf{WH}_1 + \mathbf{b}\mathbf{1}_C^\top - \mathbb{I}_C. \tag{39}$$

Notice that $\mu_{\text{single}}$ and $\mu_{\text{class}}$ are the two largest eigenvalues of the block-structured kernel $\Theta^h_{s,s}(X)$ (see Table 1 for the eigndecomposition of a block-structured matrix), and $\mathbf{R}_1$ is a matrix of the stacked class-mean residuals, which is also defined in (17). The the dynamics (38) simplifies to

$$(40) \quad \begin{cases} \dot{\mathbf{H}}_1 = & -\mathbf{W}^\top(\mu_{\text{class}}\mathbf{R}_1 + \kappa_n N(\frac{1}{C}\mathbf{WH}_1\mathbf{1}_C\mathbf{1}_C^\top + \mathbf{b}\mathbf{1}_C^\top - \frac{1}{C}\mathbf{1}_C\mathbf{1}_C^\top)) \\ \dot{\mathbf{H}}_2 = & -\mu_{\text{single}}\mathbf{W}^\top\mathbf{WH}_2 \\ \dot{\mathbf{W}} = & -m(\mathbf{R}_1\mathbf{H}_1^\top - \mathbf{WH}_2\mathbf{H}_2^\top) \\ \dot{\mathbf{b}} = & -m\mathbf{R}_1\mathbf{1}_C. \end{cases}$$

It remains to simplify the expression for $\dot{\mathbf{H}}_1$. By using the relation

$$\frac{1}{C}\mathbf{W}\mathbf{H}_1\mathbf{1}_C\mathbf{1}_C^\top + \mathbf{b}\mathbf{1}_C^\top - \frac{1}{C}\mathbf{1}_C\mathbf{1}_C^\top = \frac{1}{C}\mathbf{R}_1\mathbf{1}_C\mathbf{1}_C^\top, \tag{41}$$

we can deduce that the dynamics for $\dot{\mathbf{H}}_1$ in (40) can be expressed as (recalling that $N = mC$)

$$\dot{\mathbf{H}}_1 = -\mathbf{W}^\top\mathbf{R}_1(\mu_{\text{class}}\mathbb{I} + \kappa_n m\mathbf{1}_C\mathbf{1}_C^\top). \tag{42}$$

We notice that $(\mathbb{I}_C + \frac{\kappa_n m}{\mu_{\text{class}}}\mathbf{1}_C\mathbf{1}_C^\top)^{-1} = \mathbb{I}_C - \alpha\mathbf{1}_C\mathbf{1}_C^\top$, where $\alpha := \frac{\kappa_n m}{\mu_{\text{class}}+C\kappa_n m}$. Then we can derive the invariant of the training dynamics by direct computation of the time-derivative $\dot{\mathbf{E}}$, where

$$\mathbf{E} := \frac{1}{m}\mathbf{W}^\top\mathbf{W} - \frac{1}{\mu_{\text{class}}}\mathbf{H}_1(\mathbb{I}_C - \alpha\mathbf{1}_C\mathbf{1}_C^\top)\mathbf{H}_1^\top - \frac{1}{\mu_{\text{single}}}\mathbf{H}_2\mathbf{H}_2^\top \tag{43}$$

Since $\dot{\mathbf{E}} = \mathbb{O}$, we get that the quantity $\mathbf{E}$ remains constant in time. This completes the proof.

$\square$

## B.3 Proof of Theorem 5.2

We divide the proof into two main parts: the first one shows the emergence of NC1, and the second one shows NC2-4.

*(NC1).* Following the analysis in Section 3, the dynamics eventually enters the end of training phase (see Section 4.1). Then the dynamics in Theorem 5.1 simplifies to the following form:

$$\begin{cases} \dot{\mathbf{H}}_1 = \mathbb{O} \\ \dot{\mathbf{H}}_2 = -\mu_{\text{single}}\mathbf{W}^\top\mathbf{W}\mathbf{H}_2 \\ \dot{\mathbf{W}} = -m\mathbf{W}\mathbf{H}_2\mathbf{H}_2^\top \\ \dot{\mathbf{b}} = \mathbb{O} \end{cases} \tag{44}$$

As we note in Section 4, this dynamics is similar to the gradient flow of the unconstrained features models and is an instance of the class of hyperbolic dynamics, which is discussed in Section 5.2. During this phase the quantity

$$\tilde{\mathbf{E}} := \mu_{\text{single}}\mathbf{W}^\top\mathbf{W} - m\mathbf{H}_2\mathbf{H}_2^\top = m\mu_{\text{single}}(\mathbf{E} + \frac{1}{\mu_{\text{class}}}\mathbf{H}_1(\mathbb{I} - \alpha\mathbf{1}_C\mathbf{1}_C^\top)\mathbf{H}_1^\top) \tag{45}$$

does not change in time. Hence we can decouple the dynamic using the invariant as follows:

$$\begin{cases} \dot{\mathbf{H}}_2 = -\mu_{\text{single}}(\tilde{\mathbf{E}} + m\mathbf{H}_2\mathbf{H}_2^\top)\mathbf{H}_2 \\ \dot{\mathbf{W}} = -\mathbf{W}(\mu_{\text{single}}\mathbf{W}^\top\mathbf{W} - \tilde{\mathbf{E}}) \end{cases} \tag{46}$$

Since $\mathbf{E}$ is p.s.d (or zero, as a special case), $\tilde{\mathbf{E}}$ is p.s.d as well, and the eigendecomposition of the invariant is given by $\tilde{\mathbf{E}} = \sum_k c_k v_k v_k^\top$ for some coefficients $c_k \geq 0$ and a set of orthonormal vectors $v_k \in \mathbb{R}^n$. Then we also have $\mathbf{H}_2\mathbf{H}_2^\top = \sum_{k,l} \alpha_{kl} v_k v_l^\top$, where $\alpha_{kl}$ are symmetric (i.e. $\alpha_{kl} = \alpha_{lk}$) and $\alpha_{kk} \geq 0$ for all $k = 1, \ldots n$ (since $\mathbf{H}_2\mathbf{H}_2^\top$ is symmetric and p.s.d.). Note that coefficients $c_k$ here are constant while coefficients $\alpha_{kl}$ are time-dependent. Let us then write the dynamics for $\alpha_{kl}$ using the dynamics of $\mathbf{H}_2\mathbf{H}_2^\top$:

$$(\mathbf{H}_2\dot{\mathbf{H}}_2^\top) = -\tilde{\mathbf{E}}\mathbf{H}_2\mathbf{H}_2^\top - \mathbf{H}_2\mathbf{H}_2^\top\tilde{\mathbf{E}} - 2(\mathbf{H}_2\mathbf{H}_2^\top)^2 \tag{47}$$

Then for the elements of $\alpha$ we have:

$$\dot{\alpha}_{kl} = -\alpha_{kl}(c_k + c_l) - 2\sum_j \alpha_{kj}\alpha_{jl} \tag{48}$$

For the diagonal elements $\alpha_{kk}$, this gives:

$$\dot{\alpha}_{kk} = -2c_k\alpha_{kk} - 2\sum_j \alpha_{kj}^2 \tag{49}$$

Since $c_k \geq 0$, $\alpha_{kk} \geq 0$ and $\alpha_{kj}^2 \geq 0$, we get that

$$\alpha_{kk} \xrightarrow[t\to\infty]{} 0 \quad \forall k \tag{50}$$

And, therefore, all the non-diagonal elements also tend to zero. Thus, we get that

$$\mathbf{H}_2\mathbf{H}_2^\top \xrightarrow[t\to\infty]{} \mathbb{O} \tag{51}$$

and thus

$$\mathbf{H}_2 \xrightarrow[t\to\infty]{} \mathbb{O} \tag{52}$$

Now we notice that from the expression for $\mathbf{H}_2$ in (24) it follows that $\mathbf{H}_2 = \mathbb{O}$ implies variability collapse, since it means that all the feature vectors within the same class are equal. Indeed, $\mathbf{H}^{(c)}\tilde{\mathbf{Q}}_2 = \mathbb{O} \in \mathbb{R}^{n\times(m-1)}$ means that there is a set of $m-1$ orthogonal vectors, which are all also orthogonal to $[h_i(x_1^c),\ldots,h_i(x_m^c)]$ for any $i = 1,\ldots,n$, where $x_i^c$ are inputs from class $c$. However, there is only one vector (up to a constant) orthogonal to all the columns of $\tilde{\mathbf{Q}}_2$ in $\mathbb{R}^m$ and this vector is $\mathbf{1}_m$. Therefore, $[h_i(x_1^c),\ldots h_i(x_m^c)] = \gamma\mathbf{1}_m$ for some constant $\gamma$ for any $i = 1,\ldots,n$. Thus, we indeed have $h(x_1^c) = \cdots = h(x_m^c)$, which constitutes variability collapse within classes. $\qquad\square$

*(NC2-4)*. Set $\beta = \frac{1}{C}$. We first show that zero global feature mean implies $\mathbf{b} = \beta\mathbf{1}_C$. At the end of training, since $\mathbf{R}_1 = \mathbb{O}$, we have

$$\mathbf{W}\mathbf{H}_1 + \mathbf{b}\mathbf{1}_C^\top = \mathbb{I}_C \tag{53}$$

On the other hand, zero global mean implies $\mathbf{H}_1\mathbf{1}_C = C\langle h\rangle = \mathbb{O}$. Then multiplying (53) by $\mathbf{1}_C$ on the right, we get the desired expression for the biases. Given the zero global mean, we have

$$\frac{1}{m}\mathbf{W}^\top\mathbf{W} - \frac{1}{\mu_{\text{class}}}\mathbf{H}_1\mathbf{H}_1^\top - \frac{1}{\mu_{\text{single}}}\mathbf{H}_2\mathbf{H}_2^\top = \mathbf{E} - \frac{\alpha m C^2}{\mu_{\text{class}}}\langle h\rangle\langle h\rangle^\top = \mathbf{E} \tag{54}$$

By the proof of NC1, $\mathbf{H}_2 \to \mathbb{O}$. Together with the assumption that $\mathbf{E}$ is proportional to the limit of $\mathbf{W}^\top\mathbf{W}$ (or zero, as a special case), we obtain

$$\mu_{\text{class}}\mathbf{W}^\top\mathbf{W} - m\mathbf{H}_1\mathbf{H}_1^\top \to \gamma\mathbf{W}^\top\mathbf{W} \tag{55}$$

for some $\gamma \geq 0$. Note that since $\mathbf{H}_1\mathbf{H}_1^\top$ is p.s.d. this implies $\tilde{\lambda}_c := \mu_{\text{class}} - \gamma \geq 0$. By multiplying the left and right with appropriate factors, we have

$$\begin{cases} \mathbf{H}_1^\top(\tilde{\lambda}_c\mathbf{W}^\top\mathbf{W} - m\mathbf{H}_1\mathbf{H}_1^\top)\mathbf{H}_1 \to \mathbb{O} \\ \mathbf{W}(\tilde{\lambda}_c\mathbf{W}^\top\mathbf{W} - m\mathbf{H}_1\mathbf{H}_1^\top)\mathbf{W}^\top \to \mathbb{O}. \end{cases} \tag{56}$$

Consequently (according to (53))

$$\begin{cases} \tilde{\lambda}_c(\mathbb{I}_C - \beta\mathbf{1}_C\mathbf{1}_C^\top)^2 - m(\mathbf{H}_1^\top\mathbf{H}_1)^2 \to \mathbb{O} \\ \tilde{\lambda}_c(\mathbf{W}\mathbf{W}^\top)^2 - (\mathbb{I}_C - \beta\mathbf{1}_C\mathbf{1}_C^\top)^2 \to \mathbb{O} \end{cases} \tag{57}$$

Since both $\mathbf{W}\mathbf{W}^\top$ and $\mathbf{H}_1^\top\mathbf{H}_1$ are p.s.d., we have

$$\begin{cases} \mathbf{H}_1^\top\mathbf{H}_1 \to \sqrt{\dfrac{\tilde{\lambda}_c}{m}}(\mathbb{I}_C - \beta\mathbf{1}_C\mathbf{1}_C^\top) \\ \mathbf{W}\mathbf{W}^\top \to \sqrt{\dfrac{m}{\tilde{\lambda}_c}}(\mathbb{I}_C - \beta\mathbf{1}_C\mathbf{1}_C^\top). \end{cases} \tag{58}$$

To establish NC2, recall that $\mathbf{H}_1 = [\langle h\rangle_1,\ldots,\langle h\rangle_C]$ and that $\mathbf{M}$, as a normalized version of $\mathbf{H}_1$, satisfies

$$\mathbf{M}^\top\mathbf{M} \to \frac{1}{1-\beta}(\mathbb{I}_C - \beta\mathbf{1}_C\mathbf{1}_C^\top) = \frac{C}{C-1}\left(\mathbb{I}_C - \frac{1}{C}\mathbf{1}_C\mathbf{1}_C^\top\right).$$

To establish NC3, note that from (55) and (58) together, it follows that the limits of $\mathbf{M}$ and $\mathbf{W}^\top$ only differ by a constant multiplier.

To establish NC4, note that using NC3 we can write

$$\operatorname*{argmax}_c(\mathbf{W}h(x) + \mathbf{b})_c = \operatorname*{argmax}_c(\mathbf{W}h(x))_c \qquad\qquad (\mathbf{b} = \beta\mathbf{1}_C)$$

$$\to \operatorname*{argmax}_c(\mathbf{M}^\top h(x))_c \qquad\qquad (\text{NC3})$$

$$= \operatorname*{argmin}_c \|h(x) - \langle h\rangle_c\|_2.$$

This completes the proof. $\qquad\square$

## B.4 General biases case

*Proof.* As in the proof of Theorem 5.2, at the end of training we have $\mathbf{W}\mathbf{H}_1 + \mathbf{b}\mathbf{1}_C^\top = \mathbb{I}_C$. Moreover, since $\mathbf{E} = \mathbb{O}$ and $\mathbf{H}_2 \to \mathbb{O}$, we have

$$\frac{1}{m}\mathbf{W}^\top\mathbf{W} - \frac{1}{\mu_{\text{class}}}\mathbf{H}_1(\mathbb{I} - \alpha\mathbf{1}_C\mathbf{1}_C^\top)\mathbf{H}_1^\top \to \mathbb{O}. \tag{59}$$

Multyplying the above expression to the left by $\mathbf{W}$ and to the right by $\mathbf{W}^\top$, we obtain the general expression (29) for the matrix $(\mathbf{W}\mathbf{W}^\top)^2$ mentioned in the main text:

$$(\mathbf{W}\mathbf{W}^\top)^2 \to \frac{m}{\mu_{\text{class}}}\Big(\mathbb{I}_C - \alpha\mathbf{1}_C\mathbf{1}_C^\top + (1 - \alpha C)(C\mathbf{b}\mathbf{b}^\top - \mathbf{b}\mathbf{1}_C^\top - \mathbf{1}_C\mathbf{b}^\top)\Big). \tag{60}$$

This expression implies that the rows of the weights matrix may have varying separation angles in the general biases case, i.e., there is no symmetric structure is general. However, for constant biases $\mathbf{b} = \beta\mathbf{1}_C$, the above expression simplifies to

$$(\mathbf{W}\mathbf{W}^\top)^2 \to \frac{m}{\mu_{\text{class}}}\Big(\mathbb{I}_C - \frac{1}{C}\big(1 - (1 - \alpha C)(1 - \beta C)^2\big)\mathbf{1}_C\mathbf{1}_C^\top\Big). \tag{61}$$

Since $\alpha < 1/C$ and $(1-\beta C)^2 \geq 0$, we have that $(1-(1-\alpha C)(1-\beta C)^2)/C \leq 1/C$. Therefore, the RHS of (61) is always p.s.d. and has a unique p.s.d square root proportional to $\mathbb{I}_C - \gamma\mathbf{1}_C\mathbf{1}_C^\top$ for some constant $\gamma < 1/C$. Denote $\rho := (1-(1-\alpha C)(1-\beta C)^2)/C$, then we have $\gamma = (1 - \sqrt{1 - C\rho})/C$. Note that $\rho < 1/C$ ensures that $\gamma$ is well defined. Then the configuration of the final weights is given by

$$\mathbf{W}\mathbf{W}^\top \to \sqrt{\frac{m}{\mu_{\text{class}}}}\Big(\mathbb{I}_C - \gamma\mathbf{1}_C\mathbf{1}_C^\top\Big). \tag{62}$$

This means that the norms of all the weights rows are still equal, as in NC2. However, since $\gamma < 1/C$ if $\beta \neq 1/C$, the angle between these rows is smaller than in the ETF structure.

We can derive the configuration of the class means similarly by multyplying (59) to the left by $\mathbf{H}_1^\top$ and to the right by $\mathbf{H}_1$. In the general biases case, we get

$$\mathbf{H}_1^\top\mathbf{H}_1(\mathbb{I}_C - \alpha\mathbf{1}_C\mathbf{1}_C^\top)\mathbf{H}_1^\top\mathbf{H}_1 \to \frac{\mu_{\text{class}}}{m}\Big(\mathbb{I}_C - \mathbf{b}\mathbf{1}_C^\top - \mathbf{1}_C\mathbf{b}^\top + \|\mathbf{b}\|_2^2\mathbf{1}_C\mathbf{1}_C^\top\Big). \tag{63}$$

As with the weights, we see that this is not a symmetric structure in general. Thus, NC2 does not hold in the general biases case. However, for the constant biases $\mathbf{b} = \beta\mathbf{1}_C$, the above expression simplifies to

$$\mathbf{H}_1^\top\mathbf{H}_1(\mathbb{I}_C - \alpha\mathbf{1}_C\mathbf{1}_C^\top)\mathbf{H}_1^\top\mathbf{H}_1 \to \frac{\mu_{\text{class}}}{m}(\mathbb{I}_C - \beta\mathbf{1}_C\mathbf{1}_C^\top)^2. \tag{64}$$

Analogously to the previous derivations, we get that the unique p.s.d. square root of the RHS is given by $\mathbb{I}_C - \tilde{\rho}\mathbf{1}_C\mathbf{1}_C^\top$, where $\tilde{\rho} := (1 - |1 - \beta C|)/C < 1/C$ for $\beta \neq 1/C$. On the other hand, the unique p.s.d root of $\mathbb{I} - \alpha\mathbf{1}_C\mathbf{1}_C^\top$ is given by $\mathbb{I}_C - \phi\mathbf{1}_C\mathbf{1}_C^\top$, where $\phi := (1 - \sqrt{1 - \alpha C})/C$. Thus, we have the following

$$\sqrt{\frac{m}{\mu_{\text{class}}}}\mathbf{H}_1^\top\mathbf{H}_1(\mathbb{I}_C - \phi\mathbf{1}_C\mathbf{1}_C^\top) \to \mathbb{I}_C - \tilde{\rho}\mathbf{1}_C\mathbf{1}_C^\top. \tag{65}$$

Therefore, the structure of the last-layer features class means is given by

$$\mathbf{H}_1^\top\mathbf{H}_1 \to \sqrt{\frac{\mu_{\text{class}}}{m}}\Big(\mathbb{I}_C - \tilde{\rho}\mathbf{1}_C\mathbf{1}_C^\top\Big)\Big(\mathbb{I}_C - \frac{\phi}{1 + \phi C}\mathbf{1}_C\mathbf{1}_C^\top\Big) = \sqrt{\frac{\mu_{\text{class}}}{m}}\Big(\mathbb{I}_C - \theta\mathbf{1}_C\mathbf{1}_C^\top\Big), \tag{66}$$

where $\theta := \tilde{\rho} + \phi/(1 + \phi C) - C\tilde{\rho}\phi/(1 + \phi C) < 1/C$ for $\beta \neq 1/C$. Thus, similarly to the classifier weights $\mathbf{W}$, the last-layer features class means form a symmetric structure with equal lengths and a separation angle smaller than in the ETF. However, the centralized class means given by $\mathbf{M} = \mathbf{H}_1(\mathbb{I}_C - \mathbf{1}_C\mathbf{1}_C^T/C)$ still form the ETF structure:

$$\mathbf{M}^\top\mathbf{M} \to \sqrt{\frac{\mu_{\text{class}}}{m}}\Big(\mathbb{I}_C - \frac{1}{C}\mathbf{1}_C\mathbf{1}_C^\top\Big). \tag{67}$$

This holds since the component proportional to $\mathbf{1}_C\mathbf{1}_C^\top$ on the RHS of equation (66) lies in the kernel of the ETF matrix $(\mathbb{I}_C - \mathbf{1}_C\mathbf{1}_C^\top/C)$. Thus, we conclude that NC2 holds in case of equal biases, while NC3 does not. $\square$

**Remark on $\alpha \to 0$ case:** Simplifying the expressions for constants $\gamma$ and $\theta$, which define the angles in the configurations of the weights and the class means above, we get the following:

$$\gamma = \frac{1}{C}\big(1 - |1 - \beta C|\sqrt{1 - \alpha C}\big), \quad \theta = \frac{1}{C}\Big(1 - \frac{|1 - \beta C|}{2 - \sqrt{1 - \alpha C}}\Big). \tag{68}$$

Analyzing these expressions, we find that they are equal only if $1 - \alpha C = 1$, i.e. $\alpha = 0$. However, this can only hold if $\kappa_n = 0$ by definition of $\alpha$, i.e., when the kernel $\Theta^h$ is zero on pairs of samples from different classes. While $\alpha \neq 0$ in general, there are certain settings where $\alpha$ approaches zero. Simplifying the expression for $\alpha$, we can get the following

$$\alpha = \frac{1}{\frac{\kappa_c}{\kappa_n}(1 - \frac{1}{m}) + \frac{\kappa_d}{\kappa_n}\frac{1}{m} + (C - 1)}. \tag{69}$$

One can see that $\alpha \to 0$ if $C \to \infty$ or when $\kappa_c/\kappa_n \to \infty$. Since the kernel $\Theta^h$ is strongly aligned with the labels in our numerical experiments, the value of $\kappa_c/\kappa_n$ is large in practice. Thus, $\alpha$ is not zero but indeed significantly smaller than $1/C$. Thus, in our numerical experiments the angles $\theta$ and $\gamma$ are close to each other. However, we note that the equality of these two angles does not imply NC3, since the value of $\theta$ characterizes the angles between the non-centralized class means.

**Remark on $\alpha \to 1/C$ case:** If $\alpha = 1/C$, the equation (63) for the structure of the features class means with general (not equal) biases simplifies to

$$\mathbf{M}^\top \mathbf{M} \to \frac{\mu_{\text{class}}}{m}\Big(\mathbb{I}_C - \frac{1}{C}\mathbf{1}_C\mathbf{1}_C^\top\Big), \tag{70}$$

i.e., in this case the class means always exhibit the ETF structure, even without the assumption that all the biases are equal. Moreover, in this case $\gamma = 1/C$ as well. Thus, both NC2 and NC3 hold. While by definition $\alpha < 1/C$, we can analyze the cases when it approaches $1/C$ using the expression (69) again. One can see that when $m \to \infty$ and $\kappa_c/\kappa_n \to 1$, we have $\alpha \to 1/C$. However, the requirement $\kappa_c/\kappa_n \to 1$ implies that the kernel $\Theta^h$ does not distinguish between pairs of samples from the same class and from different classes. Such a property of the kernel is associated with poor generalization performance and does not occur in our numerical experiments.

## C   Numerical experiments

**Implementation details**   We use JAX [8] and Flax (neural network library for JAX) [25] to implement all the DNN architectures and the training routines. This choice of the software allows to compute the empirical NTK of any DNN architecture effortlestly and efficiently. We compute the values of kernels $\Theta$ and $\Theta^h$ on the whole training batch ($m = 12$ samples per class, 120 samples in total) in case of ResNet20 and DenseNet40 to approximate the values $(\gamma_d, \gamma_c, \gamma_n)$ and $(\kappa_d, \kappa_c, \kappa_n)$, as well as the NTK alignment metrics, and compute the invariant $\mathbf{E}$ using these values. Since VGG11 and VGG16 architectures are much larger (over 10 million parameters) and computing their Jacobians is very memory-intensive, we use $m = 4$ samples per class (i.e., 40 samples in total) to approximate the kernels of these models. We compute all the other training metrics displayed in panes a-e of Figures 3, 4, 5, 6, 7, 8, 9, 10, 11 on the whole last batch of every second training epoch for all the architectures. The test accuracy is computed on the whole test set. To produce panes f-h of the same figures, we only compute the NC metrics and the test accuracy one time after 400 epochs of training for every learning rate. We use 30 logarithmically spaced learning rates in the range $\eta \in [10^{-4}, 10^{0.25}]$ for ResNet20 trained on MNIST and VGG11 trained on MNIST. For all the other architecture-dataset pairs we only compute the last 20 of these learning rates to reduce the computational costs, since the smallest learning rates do not yield models with acceptable performance.

**Compute**   We executed the numerical experiments mainly on NVIDIA GeForce RTX 3090 Ti GPUs, each model was trained on a single GPU. In this setup, a single training run displayed in panes a-e of Figures 3, 4, 5, 6, 7, 8, 9, 10, 11 took approximately 3 hours for ResNet20, 6 hours for DenseNet40, 7 hours for VGG11, and 11 hours for VGG16. This adds up to a total of 312 hours to compute panes a-e of the figures. The computation time is mostly dedicated not to the training routine itself but to the large number of computationally-heavy metrics, which are computed every second epoch of a training run. Indeed, to approximate the values of $\Theta$ and $\Theta^h$, one needs to compute $C(C + 1) + n(n + 1)$ kernels on a sample of size $mC$ from the dataset, and each of the kernels requires computing a

gradient with respect to numerous parameters of a DNN. Additionally, the graphs in panes f-h of the same figures take around 1.5 hours for each learning rate value for ResNet20, 3 hours for DenseNet40, and 4 hours for VGG11 and VGG16, which adds up to approximately 1350 computational hours.

**Results**   We include experiments on the following architecture-dataset pairs:

- Figure 3: VGG11 trained on MNIST
- Figure 4: VGG11 trained on FashionMNIST
- Figure 5: VGG16 trained on CIFAR10
- Figure 6: ResNet20 trained on MNIST
- Figure 7: ResNet20 trained on FashionMNIST
- Figure 8: ResNet20 trained on CIFAR10
- Figure 9: DenseNet40 trained on MNIST
- Figure 10: DenseNet40 trained on FashionMNIST
- Figure 11: DenseNet40 trained on CIFAR10

The experiments setup is described in Section 6. Panes a-h of Figures 3, 4, 5, 6, 7, 8, 9, 10, 11 are analogous to the same panes of Figure 2. We include additional pane i here, which displays the norms of the invariant terms corresponding to the feature matrix components $\mathbf{H}_1$ and $\mathbf{H}_2$, and the global features mean $\langle h \rangle$ at the end of training. One can see that the global features mean is relatively small in comparison with the class-means in every setup, and the "variance" term $\mathbf{H}_2$ is small for models that exhibit NC. We also add pane j, which displays the alignment of kernels $\Theta$ and $\Theta^h$ for every model at the end of training. One can see that the kernel alignments is typically stronger in models that exhibit NC.

## C.1   Additional examples of the NTK block structure

We include the following additional illustrative figures (analogous to Figure 1 in the main text) that show the NTK block structure in dataset-architecture pairs covered in our experiments:

- Figure 13: VGG11 trained on MNIST
- Figure 14: VGG11 trained on FashionMNIST
- Figure 15: VGG16 trained on CIFAR10
- Figure 16: ResNet20 trained on FashionMNIST
- Figure 17: ResNet20 trained on CIFAR10
- Figure 18: DenseNet40 trained on MNIST
- Figure 19: DenseNet40 trained on FashionMNIST
- Figure 11: DenseNet40 trained on CIFAR10

Overall, the block structure pattern is visible in the traced kernels in all the figures. As expected, the block structure is more pronounced in the kernels where the final alignment values are higher. While the norms of the "non-diagonal" components of the kernels are generally smaller than the "diagonal" components in panes c) and d), we notice that there is a large variability in the norms of the "diagonal" components in some settings. This means that different neurons of the penultimate layer and different classification heads may contribute to the kernel unequally in some settings. Moreover, certain "non-diagonal" components of the last-layer kernel may have non-negligible effect in some settings. We discuss how one could generalize our analysis to account for these properties of the NTK in Appendix D.

## C.2   Preliminary experiments with CE loss

While CE loss is a common choice for training DNN classifiers, our theoretical analysis and the experimental results only cover DNNs trained with MSE loss. For completeness, we provide experimental results for ResNet20 trained on MNIST with CE loss in Figure 12. One can see that

smaller invariant norm and higher invariant alignment correlate with NC in the figure. However, DNNs trained with CE loss overall reach better NC metrics but have much larger norm of the invariant in comparison with DNNs trained with MSE loss.

# D   Relaxation of the NTK Block-Structure Assumption

In this section, we first derive the dynamics equations of DNNs with a general block structure assumption on the last-layer kernel $\Theta^h$ (analogous to the equations presented in Theorem 4.1 and Theorem 5.1). Then we discuss a possible relaxation of Assumption 3.2, under which our main result regarding NC in Theorem 5.2 still holds.

## D.1   Dynamics under General Block Structure Assumption

We first formulate the most general form of the block structure assumption on $\Theta^h$ as follows:

**Assumption D.1.** *Assume that $\Theta^h : \mathcal{X} \times \mathcal{X} \to \mathbb{R}^{n \times n}$ has the following block structure*

$$\Theta^h(x, x) = \mathbf{A}_d + \mathbf{A}_c + \mathbf{A}_n, \quad \Theta^h(x_i^c, x_j^c) = \mathbf{A}_c + \mathbf{A}_n, \quad \Theta^h(x_i^c, x_j^{c'}) = \mathbf{A}_n, \qquad (71)$$

*where $\mathbf{A}_{d,c,n} \in \mathbb{R}^{n \times n}$ are arbitrary p.s.d. matrices. Here $x_i^c$ and $x_j^c$ are two distinct inputs from the same class, and $x_j^{c'}$ is an input from class $c' \neq c$.*

This assumption means that every kernel matrix $\Theta_{k,s}^h(X), k, s \in [1, n]$ still has at most three distinct values, corresponding to the inter-class, intra-class, and the diagonal values of the kernel. However, these values are arbitrary and may depend on the choice of $k, s \in [1, n]$.

Under the general block structure assumption, the gradient flow dynamics of DNNs with MSE loss takes the following form:

$$\begin{cases} \dot{\mathbf{H}} = & -\mathbf{A}_d \mathbf{W}^\top \mathbf{R} + m \mathbf{A}_c \mathbf{W}^\top \mathbf{R}_{\text{class}} + N \mathbf{A}_n \mathbf{W}^\top \mathbf{R}_{\text{global}} \\ \dot{\mathbf{W}} = & -\mathbf{R} \mathbf{H}^\top \\ \dot{\mathbf{b}} = & -\mathbf{R}_{\text{global}} \mathbf{1}_N. \end{cases} \qquad (72)$$

This is the generalized version of the dynamics presented in Theorem 4.1. Consequently, the decomposed dynamics presented in Theorem 5.1 takes the following form under the general block structure assumption:

$$\begin{cases} \dot{\mathbf{H}}_1 & = -(\mathbf{A}_d + m\mathbf{A}_c)\mathbf{W}^\top \mathbf{R}_1 - m\mathbf{A}_n \mathbf{W}^\top \mathbf{R}_1 \mathbf{1}_C \mathbf{1}_C^\top \\ \dot{\mathbf{H}}_2 & = -\mathbf{A}_d \mathbf{W}^\top \mathbf{W} \mathbf{H}_2 \\ \dot{\mathbf{W}} & = -m(\mathbf{R}_1 \mathbf{H}_1^\top + \mathbf{W} \mathbf{H}_2 \mathbf{H}_2^\top) \\ \dot{\mathbf{b}} & = -m \mathbf{R}_1 \mathbf{1}_C. \end{cases} \qquad (73)$$

The derivation of the above dynamics equations are identical to the proofs of Theorem 4.1 and Theorem 5.1 presented in Appendix B.

**Rotation invariance**   We notice that the dynamics of $(\mathbf{W}, \mathbf{H})$ in (72) has to be rotation invariant, i.e., the equations should not be affected by a change of variables $\mathbf{W} \to \mathbf{W}\mathbf{Q}, \mathbf{H} \to \mathbf{Q}^\top \mathbf{H}$ for any orthogonal matrix $\mathbf{Q}$. This holds since the loss function only depends on the product $\mathbf{W}\mathbf{H}$, which does not change under rotation. This requirement puts conditions on the behavior of $\mathbf{A}_{d,c,n}$ under rotation. Indeed, assume that the rotation $\mathbf{W} \to \mathbf{W}\mathbf{Q}, \mathbf{H} \to \mathbf{Q}^\top \mathbf{H}$ for some $\mathbf{Q}$ corresponds to the following change of the kernel:

$$\mathbf{A}_{d,c,n} \to \tilde{\mathbf{A}}_{d,c,n}(\mathbf{Q}), \qquad (74)$$

then the rotation invariance of the dynamics implies the following equality for any $\mathbf{Q}$:

$$\mathbf{Q}\tilde{\mathbf{A}}_d(\mathbf{Q})\mathbf{Q}^\top \mathbf{W}^\top \mathbf{R} + m\mathbf{Q}\tilde{\mathbf{A}}_c(\mathbf{Q})\mathbf{Q}^\top \mathbf{W}^\top \mathbf{R}_{\text{class}} + N\mathbf{Q}\tilde{\mathbf{A}}_n(\mathbf{Q})\mathbf{Q}^\top \mathbf{W}^\top \mathbf{R}_{\text{global}} \qquad (75)$$

$$= \mathbf{A}_d \mathbf{W}^\top \mathbf{R} + m\mathbf{A}_c \mathbf{W}^\top \mathbf{R}_{\text{class}} + N\mathbf{A}_n \mathbf{W}^\top \mathbf{R}_{\text{global}}. \qquad (76)$$

These equations are satisfied trivially with our initial assumption, where $\mathbf{A}_{d,c,n} = \tilde{\mathbf{A}}_{d,c,n}(\mathbf{Q}) \propto \mathbb{I}_n$. However, as we can see, any generalized assumption should specify the behavior of the kernel under rotation, and satisfy the above equation.

For general $\mathbf{A}_{d,c,n}$, the following behavior under rotation trivially satisfies the above condition: $\tilde{\mathbf{A}}_{d,c,n}(\mathbf{Q}) = \mathbf{Q}^\top \mathbf{A}_{d,c,n}\mathbf{Q}$. This behaviour of the kernel under rotation is intuitive, since it implies that the gradients of the last-layer features $h$ are rotated in the same way as the features. However, we note that gradients of parametrized functions do not in general behave this way, since the rotation of the function has to be realized by a certain change of parameters. Consider, for instance, a one-hidden-layer linear network with weights $\mathbf{V}$ in the first layer. Then we have $\mathbf{H} = \mathbf{V}X$, and a rotation $\mathbf{H} \to \mathbf{Q}^\top \mathbf{H}$ corresponds to the change of parameters $\mathbf{V} \to \mathbf{Q}^\top \mathbf{V}$. In this case, the kernel does not change under rotation, i.e., $\tilde{\mathbf{A}}_{d,c,n}(\mathbf{Q}) = \mathbf{A}_{d,c,n}$.

**Dynamics invariant** We note that the dynamics in 73 does not in general have an invariant analogous to the one we identified in Theorem 5.1. Indeed, if we define a quantity $\mathbf{E} := \mathbf{W}^\top \mathbf{W} - c_1 \mathbf{H}_1 \mathbf{H}_1^\top - c_2 \mathbf{H}_2 \mathbf{H}_2^\top$ for some constants $c_{1,2} \in \mathbb{R}$, and additionally assume centered global means $\mathbf{H}_1 \mathbf{1}_C = 0$, we get the following expression for the derivative of $\mathbf{E}$:

$$\dot{\mathbf{E}} = \left( c_1(\mathbf{A}_d + m\mathbf{A}_c) - m\mathbb{I}_n \right)\mathbf{W}^\top \mathbf{R}_1 \mathbf{H}_1^\top - \mathbf{H}_1 \mathbf{R}_1^\top \mathbf{W}\left( c_1(\mathbf{A}_d + m\mathbf{A}_c)^\top - m\mathbb{I}_n \right) \tag{77}$$

$$+ \left( c_2 \mathbf{A}_d - m\mathbb{I}_n \right)\mathbf{W}^\top \mathbf{W}\mathbf{H}_2 \mathbf{H}_2^\top - \mathbf{H}_2 \mathbf{H}_2^\top \mathbf{W}^\top \mathbf{W}\left( c_2 \mathbf{A}_d^\top - m\mathbb{I}_n \right), \tag{78}$$

which is not equal to zero with arbitrary matrices $\mathbf{A}_{d,c}$.

## D.2 Neural Collapse under Relaxed Block Structure Assumption

We now propose a relaxation of our main assumption, under which our main result regarding NC in Theorem 5.2 still holds. In terms of Assumption D.1 on the general block structure of $\Theta^h$, our initial Assumption 3.2 in the main text is the special case with $\mathbf{A}_n = \kappa_n \mathbb{I}_n$, $\mathbf{A}_c = (\kappa_c - \kappa_n)\mathbb{I}_n$, $\mathbf{A}_d = (\kappa_d - \kappa_c)\mathbb{I}_n$. The relaxed assumption can be formulated as follows in terms of matrices $\mathbf{A}_{d,c,n}$:

**Assumption D.2.** *Assume that $\mathbf{A}_n$ is an arbitrary p.s.d. matrix and $(\mathbf{A}_c, \mathbf{A}_d)$ satisfy the following conditions:*

$$\mathbf{A}_c = \kappa_c \mathbb{I}_n + \mathbf{N}_c, \quad \mathbf{A}_d = \kappa_d \mathbb{I}_n + \mathbf{N}_d, \tag{79}$$

*where $\mathbf{N}_{c,d}^\top \in \ker(\mathbf{R}^\top \mathbf{W})$, i.e., $\mathbf{N}_{c,d}\mathbf{W}^\top \mathbf{R} = \mathbb{O}$. Further, assume that the kernel changes under rotation with an orthogonal matrix $\mathbf{Q}$ as follows:*

$$\tilde{\mathbf{A}}_{d,c,n}(\mathbf{Q}) = \mathbf{Q}^\top \mathbf{A}_{d,c,n}\mathbf{Q}. \tag{80}$$

Since $\mathbf{A}_n$ is arbitrary, this relaxation allows arbitrary non-zero values of non-diagonal kernels $\Theta_{k,s}^h$ with $k \neq s$. The following observations justify the consistency of the above assumption:

- Since $\mathbf{W} \in \mathbb{R}^{C\times n}$, $\mathbf{R} \in \mathbb{R}^{C\times N}$ and $N > n > C$, $\mathbf{R}^\top \mathbf{W}$ has a non-empty kernel (possibly time-dependent).

- The dynamics is rotation invariant under the assumption, i.e., the equation (75) holds.

- The expression of the assumption is rotation invariant, in a sense that $\tilde{\mathbf{A}}_{d,c}(\mathbf{Q}) = \kappa_{d,c}\mathbb{I}_n + \tilde{\mathbf{N}}_{d,c}(\mathbf{Q})$, where $\tilde{\mathbf{N}}_{d,c}^\top(\mathbf{Q}) \in \ker(\mathbf{R}^\top \mathbf{W}\mathbf{Q})$ for any orthogonal $\mathbf{Q}$.

Under the above assumption, the derivative in 77 becomes zero, so the dynamics has an invariant of the form $\mathbf{E} := \mathbf{W}^\top \mathbf{W} - c_1 \mathbf{H}_1 \mathbf{H}_1^\top - c_2 \mathbf{H}_2 \mathbf{H}_2^\top$. Moreover, the statement and the proof of our main Theorem 5.2 remains unchanged. Thus, DNNs satisfying the conditions of Theorem 5.2 display NC under Assumption D.2.

## D.3 Discussion

The analysis of the DNNs dynamics is simplified significantly by assuming that $\Theta^h$ has a block structure. However, formulating a reasonable and consistent assumption on the NTK and its components is non-trivial. The Assumption 3.2 that we used in the main text is justified by the empirical results but may not capture all the relevant properties of the NTK. We believe that studying DNNs' dynamics under a more general or a more reasonable assumption on the NTK is a promising future work direction. The relaxed block structure assumption proposed in this section is the first step into this direction.

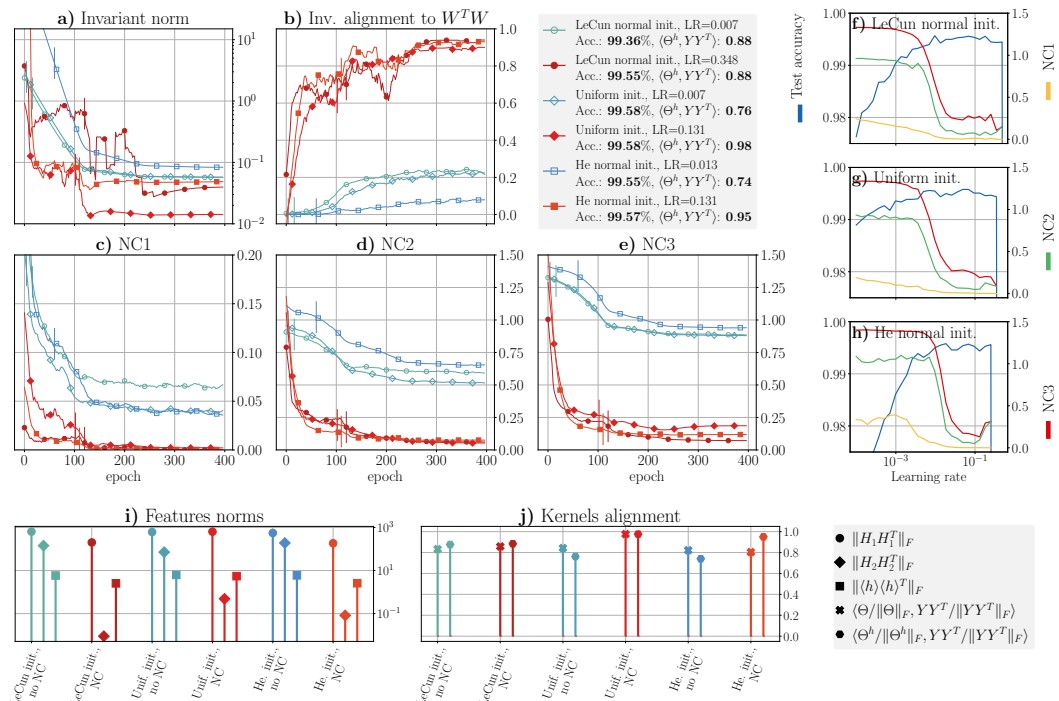

Figure 3: VGG11 trained on MNIST. See Figure 2 for the description of panes a-h. **i)** Norms of matrices $\mathbf{H}_1\mathbf{H}_1^\top$, $\mathbf{H}_2\mathbf{H}_2^\top$, and $\langle h \rangle \langle h \rangle^\top$ at the end of training. **j)** Alignment of kernels $\Theta$ and $\Theta^h$ at the end of training. The color in panes i-j is the color of the same model in panes a-e.

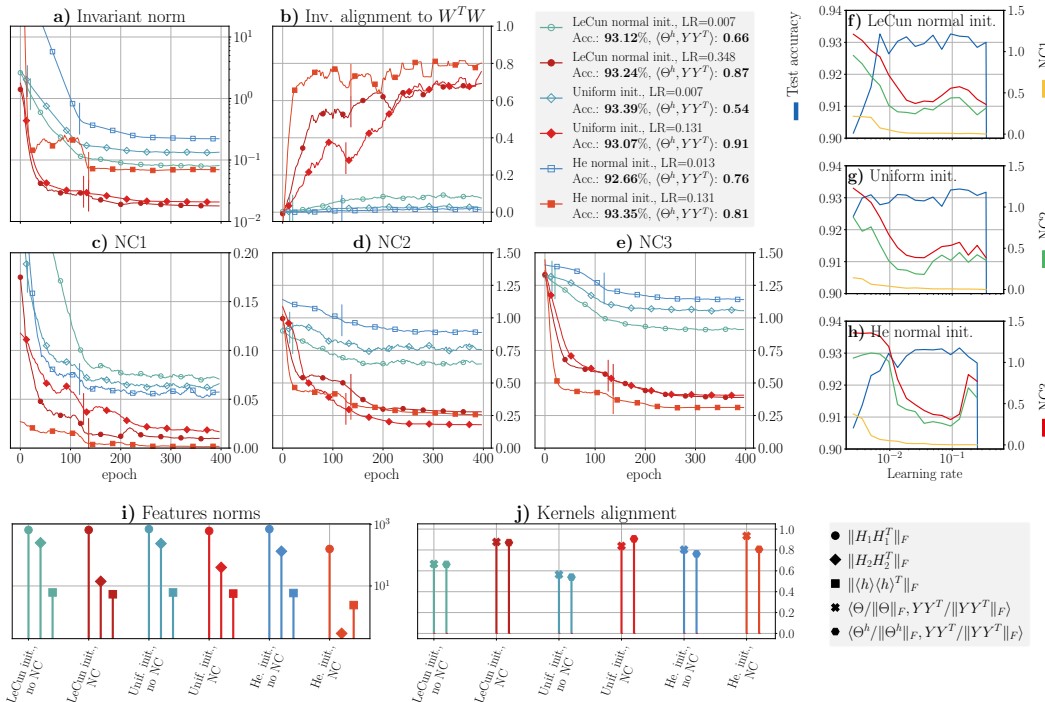

Figure 4: VGG11 trained on FashionMNIST. See Figure 2 for the description of panes a-h. **i)** Norms of matrices $\mathbf{H}_1\mathbf{H}_1^\top$, $\mathbf{H}_2\mathbf{H}_2^\top$, and $\langle h \rangle \langle h \rangle^\top$ at the end of training. **j)** Alignment of kernels $\Theta$ and $\Theta^h$ at the end of training. The color in panes i-j is the color of the same model in panes a-e.

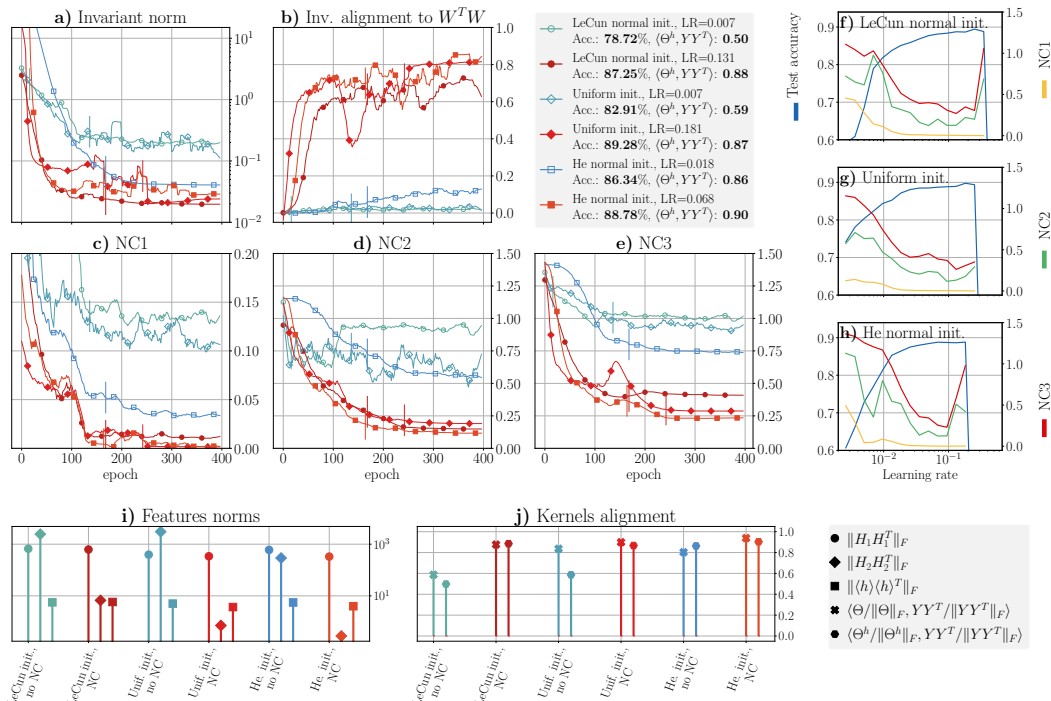

Figure 5: VGG16 trained on CIFAR10. See Figure 2 for the description of panes a-h. **i)** Norms of matrices $\mathbf{H}_1\mathbf{H}_1^\top$, $\mathbf{H}_2\mathbf{H}_2^\top$, and $\langle h \rangle \langle h \rangle^\top$ at the end of training. **j)** Alignment of kernels $\Theta$ and $\Theta^h$ at the end of training. The color in panes i-j is the color of the same model in panes a-e.

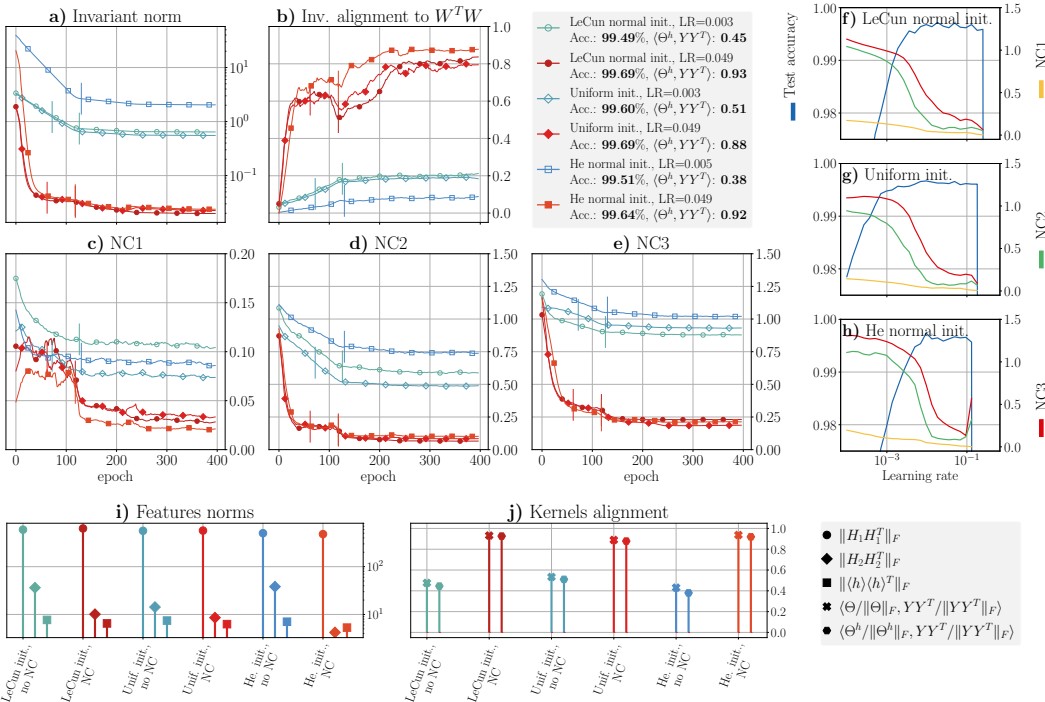

Figure 6: ResNet20 trained on MNIST. See Figure 2 for the description of panes a-h. **i)** Norms of matrices $\mathbf{H}_1\mathbf{H}_1^\top$, $\mathbf{H}_2\mathbf{H}_2^\top$, and $\langle h \rangle \langle h \rangle^\top$ at the end of training. **j)** Alignment of kernels $\Theta$ and $\Theta^h$ at the end of training. The color in panes i-j is the color of the same model in panes a-e.

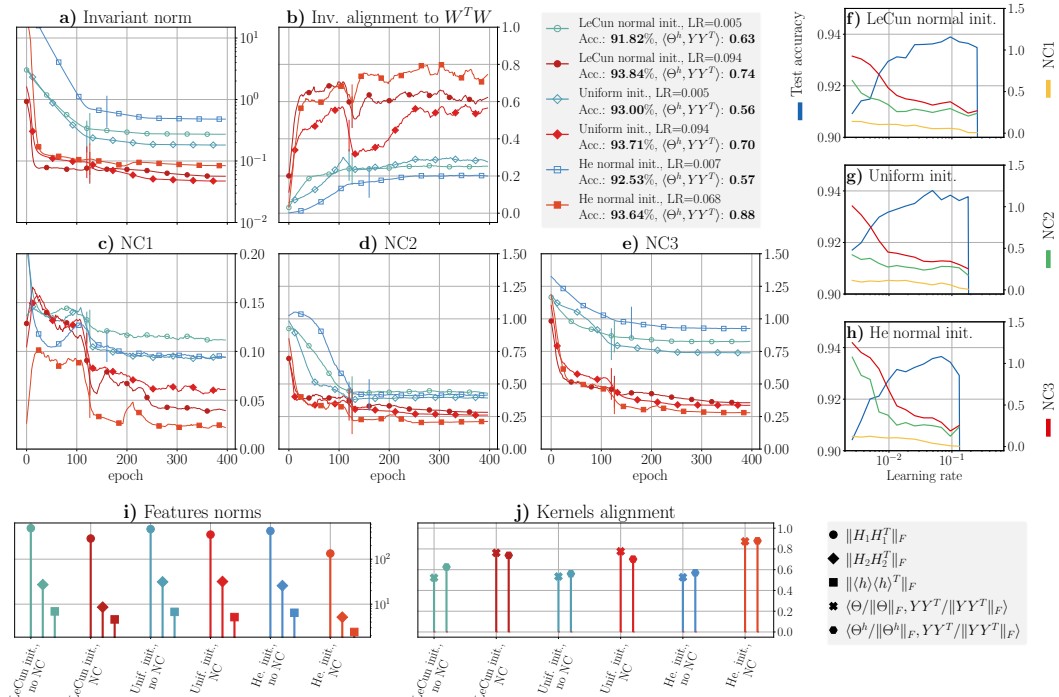

Figure 7: ResNet20 trained on FashionMNIST. See Figure 2 for the description of panes a-h. **i)** Norms of matrices $\mathbf{H}_1\mathbf{H}_1^\top$, $\mathbf{H}_2\mathbf{H}_2^\top$, and $\langle h \rangle \langle h \rangle^\top$ at the end of training. **j)** Alignment of kernels $\Theta$ and $\Theta^h$ at the end of training. The color in panes i-j is the color of the same model in panes a-e.

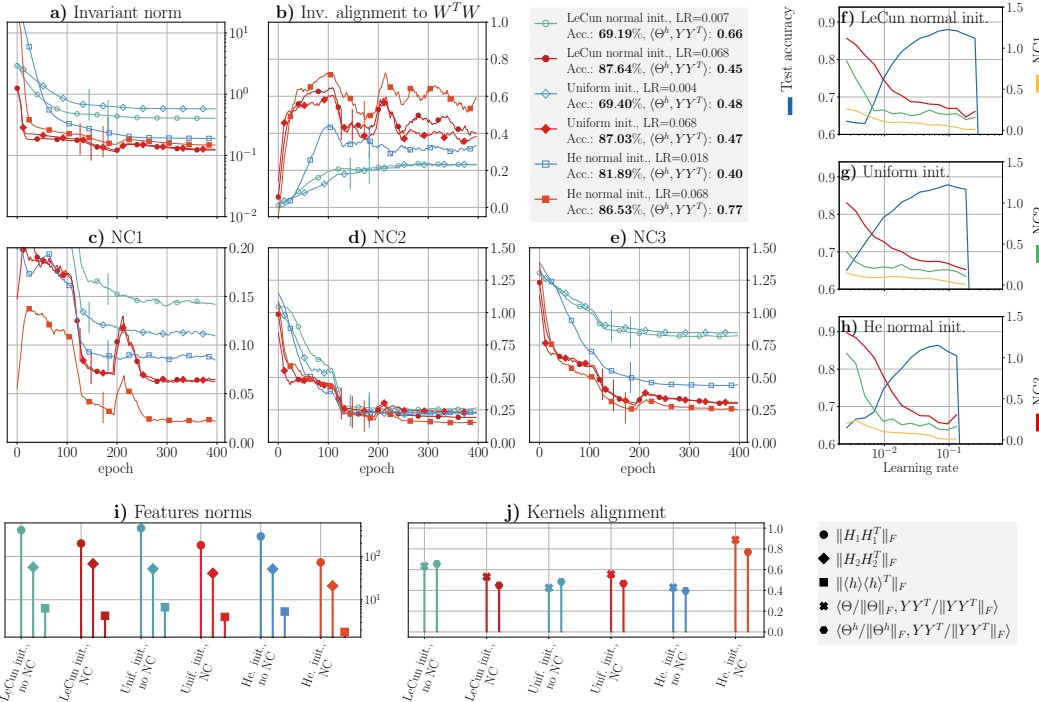

Figure 8: ResNet20 trained on CIFAR10. See Figure 2 for the description of panes a-h. **i)** Norms of matrices $\mathbf{H}_1\mathbf{H}_1^\top$, $\mathbf{H}_2\mathbf{H}_2^\top$, and $\langle h \rangle \langle h \rangle^\top$ at the end of training. **j)** Alignment of kernels $\Theta$ and $\Theta^h$ at the end of training. The color in panes i-j is the color of the same model in panes a-e.

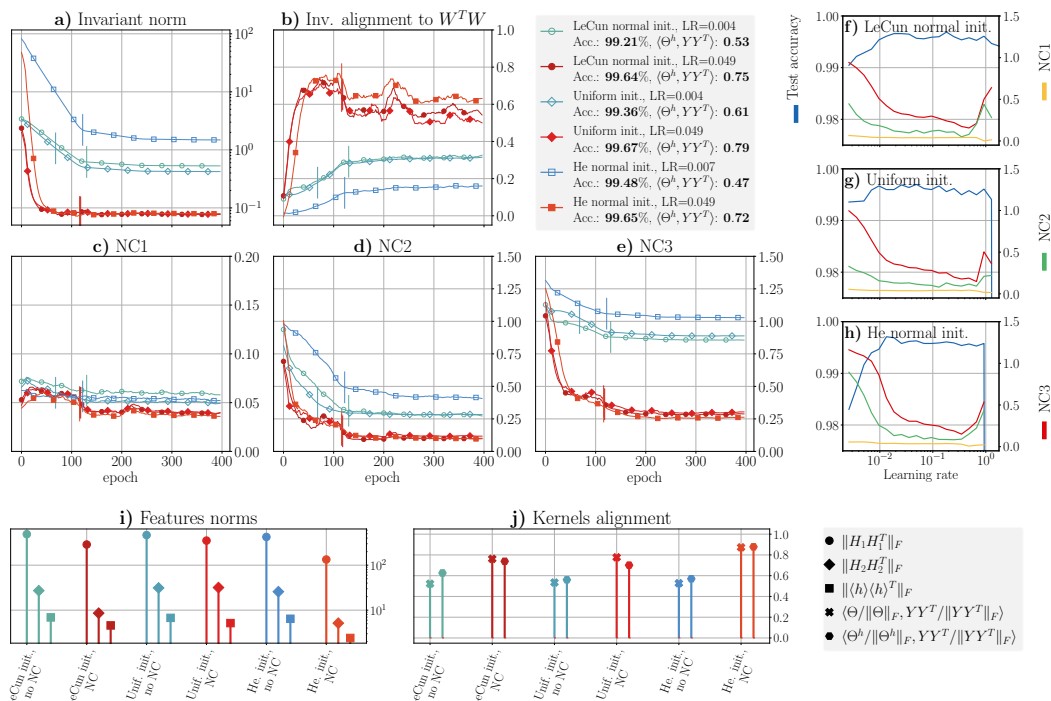

Figure 9: DenseNet40 trained on MNIST. See Figure 2 for the description of panes a-h. **i)** Norms of matrices $\mathbf{H}_1\mathbf{H}_1^\top$, $\mathbf{H}_2\mathbf{H}_2^\top$, and $\langle h \rangle \langle h \rangle^\top$ at the end of training. **j)** Alignment of kernels $\Theta$ and $\Theta^h$ at the end of training. The color in panes i-j is the color of the same model in panes a-e.

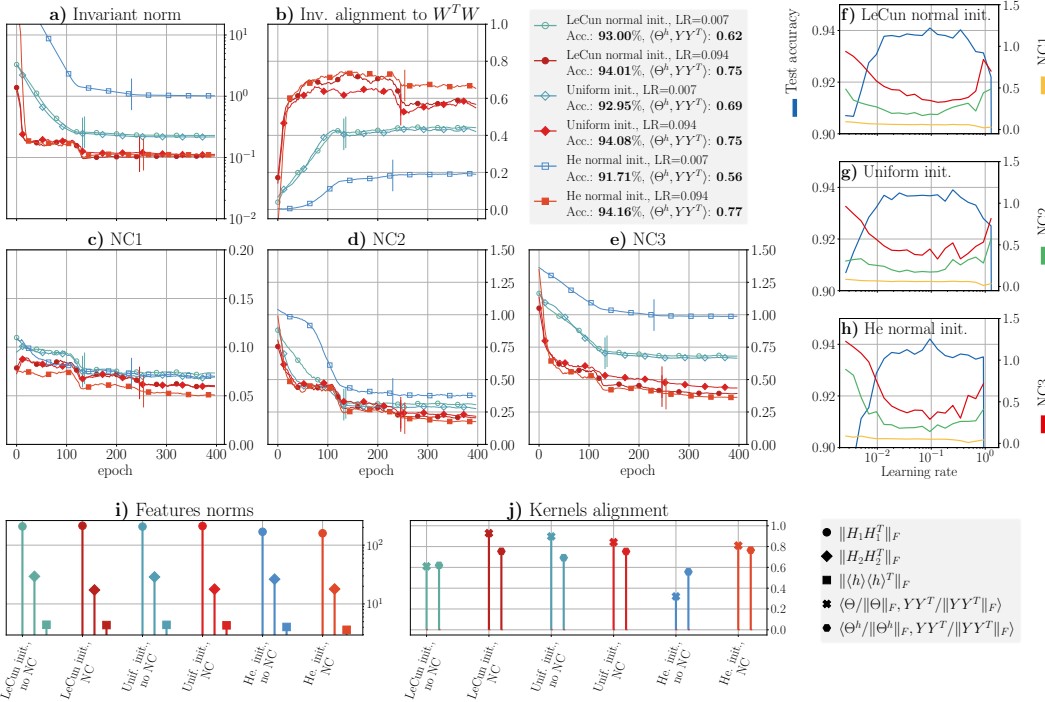

Figure 10: DenseNet40 trained on FashionMNIST. See Figure 2 for the description of panes a-h. **i)** Norms of matrices $\mathbf{H}_1\mathbf{H}_1^\top$, $\mathbf{H}_2\mathbf{H}_2^\top$, and $\langle h \rangle \langle h \rangle^\top$ at the end of training. **j)** Alignment of kernels $\Theta$ and $\Theta^h$ at the end of training. The color in panes i-j is the color of the same model in panes a-e.

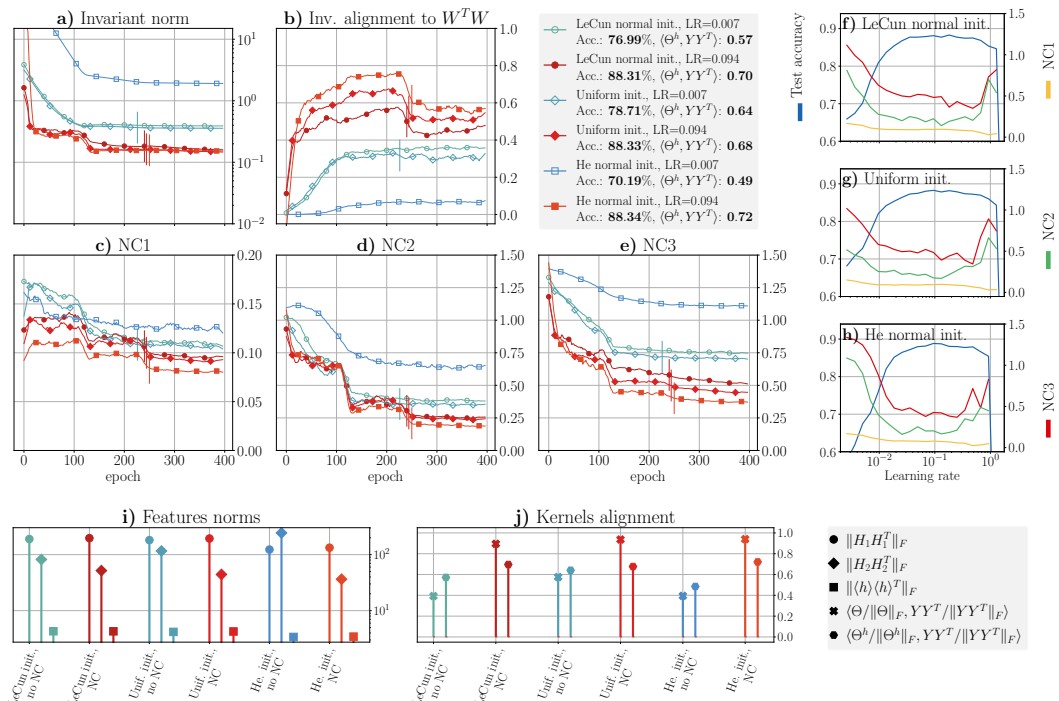

Figure 11: DenseNet40 trained on CIFAR10. See Figure 2 for the description of panes a-h. **i)** Norms of matrices $\mathbf{H}_1\mathbf{H}_1^\top$, $\mathbf{H}_2\mathbf{H}_2^\top$, and $\langle h \rangle \langle h \rangle^\top$ at the end of training. **j)** Alignment of kernels $\Theta$ and $\Theta^h$ at the end of training. The color in panes i-j is the color of the same model in panes a-e.

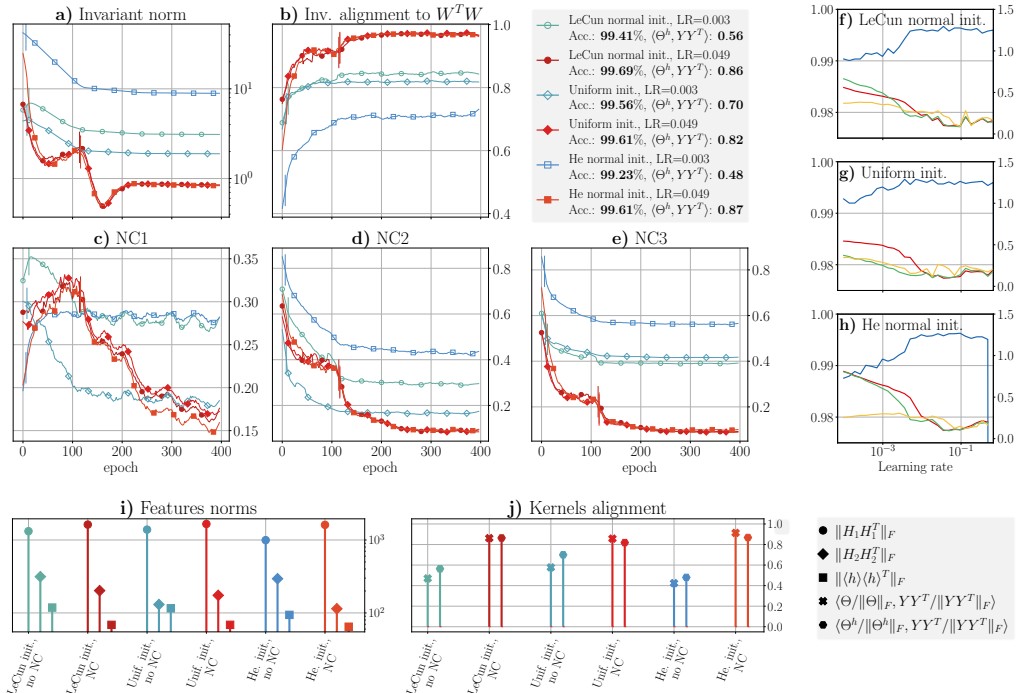

Figure 12: ResNet20 trained on MNIST with *CE loss*. See Figure 2 for the description of panes a-h. **i)** Norms of matrices $\mathbf{H}_1\mathbf{H}_1^\top$, $\mathbf{H}_2\mathbf{H}_2^\top$, and $\langle h \rangle \langle h \rangle^\top$ at the end of training. **j)** Alignment of kernels $\Theta$ and $\Theta^h$ at the end of training. The color in panes i-j is the color of the same model in panes a-e.

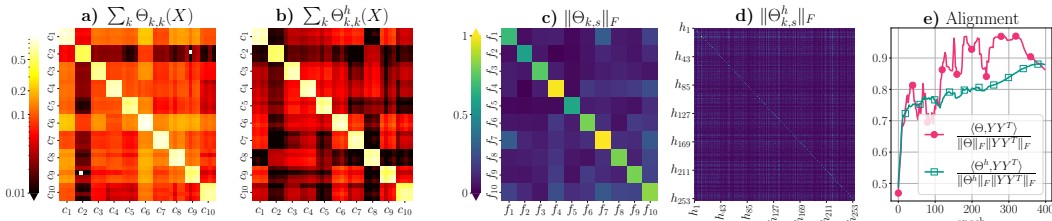

Figure 13: NTK block structure of VGG11 trained on MNIST. LeCun normal initialization, initial learning rate $0.131$. The kernel is computed on a random data subset with 4 samples from each class. See Figure 1 for the description of panes.

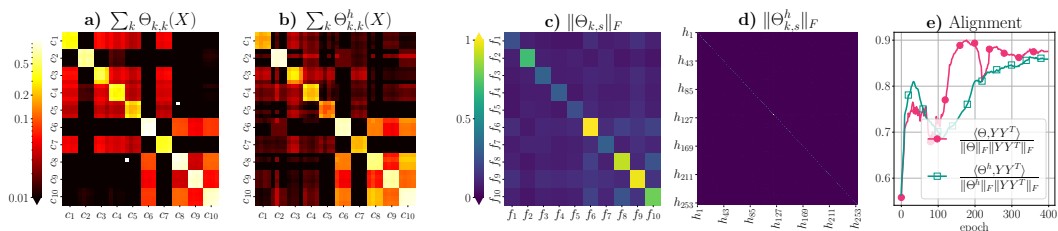

Figure 14: NTK block structure of VGG11 trained on FashionMNIST. LeCun normal initialization, initial learning rate $0.049$. The kernel is computed on a random data subset with 4 samples from each class. See Figure 1 for the description of panes.

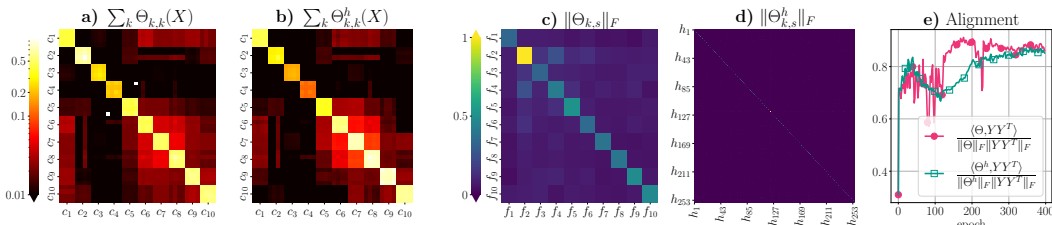

Figure 15: NTK block structure of VGG11 trained on CIFAR10. LeCun normal initialization, initial learning rate $0.131$. The kernel is computed on a random data subset with 4 samples from each class. See Figure 1 for the description of panes.

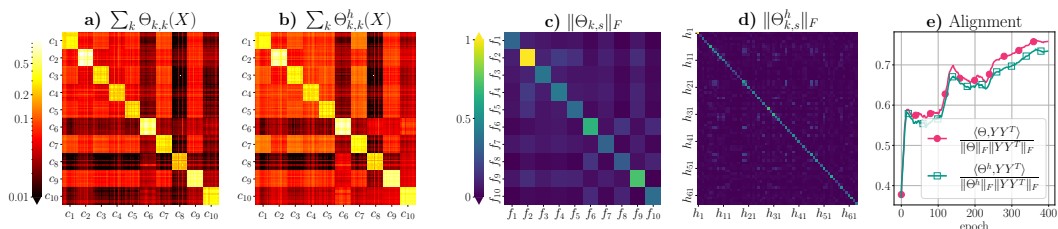

Figure 16: NTK block structure of ResNet20 trained on FashionMNIST. LeCun normal initialization, initial learning rate $0.094$. The kernel is computed on a random data subset with 12 samples from each class. See Figure 1 for the description of panes.

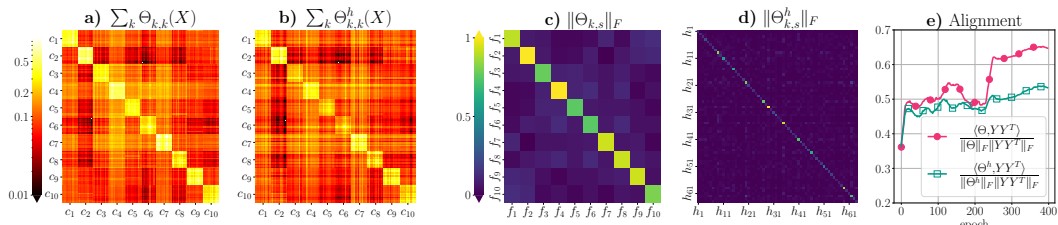

Figure 17: NTK block structure of ResNet20 trained on CIFAR10. LeCun normal initialization, initial learning rate $0.068$. The kernel is computed on a random data subset with 12 samples from each class. See Figure 1 for the description of panes.

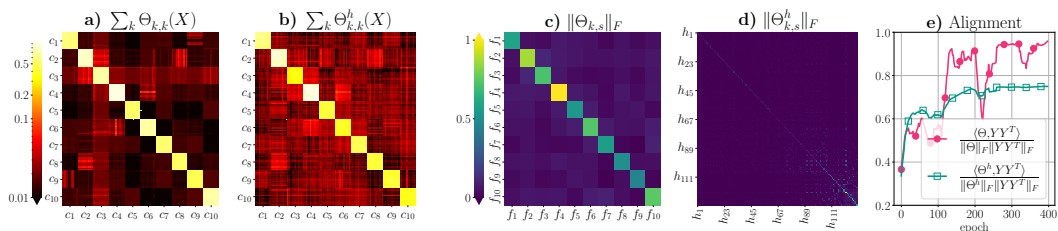

Figure 18: NTK block structure of DenseNet40 trained on MNIST. LeCun normal initialization, initial learning rate $0.049$. The kernel is computed on a random data subset with 12 samples from each class. See Figure 1 for the description of panes.

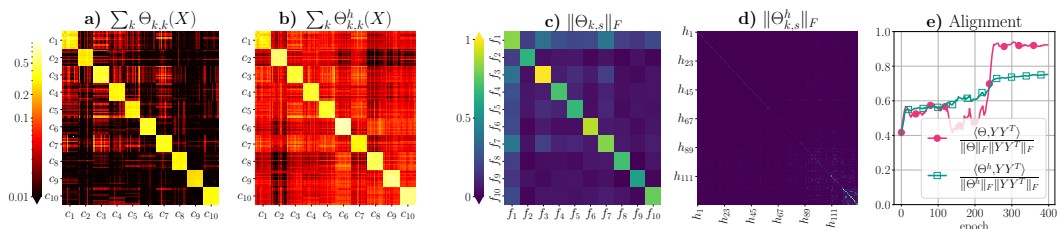

Figure 19: NTK block structure of DenseNet40 trained on FashionMNIST. LeCun normal initialization, initial learning rate $0.094$. The kernel is computed on a random data subset with 12 samples from each class. See Figure 1 for the description of panes.

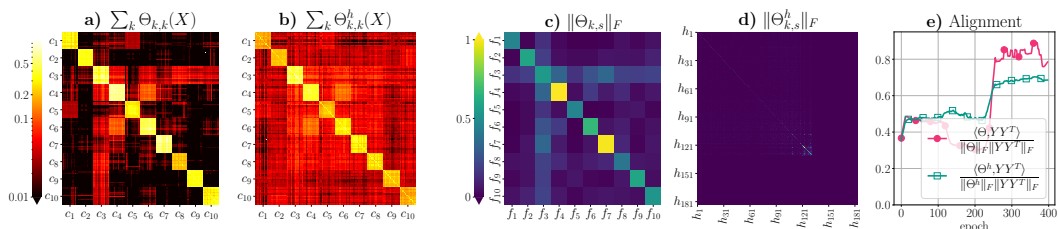

Figure 20: NTK block structure of DenseNet40 trained on CIFAR10. LeCun normal initialization, initial learning rate $0.094$. The kernel is computed on a random data subset with 12 samples from each class. See Figure 1 for the description of panes.

