# OpenReview forum: "Neural (Tangent Kernel) Collapse"
_NeurIPS.cc/2023/Conference — NeurIPS 2023 poster_

### Official Review · Reviewer_MvSr · 2023-07-02

**Soundness:** 3 good
**Presentation:** 4 excellent
**Contribution:** 2 fair
**Rating:** 6
**Confidence:** 4

**Summary:**

Previous work has observed an increased alignment and emergence of an approximate block structure in a trained network's Neural Tangent Kernel (NTK) as well as the Neural Collapse (NC) phenomenon in the last hidden layer. The paper attempts to connect the two by showing that in an extreme case of perfect block structure in NTK, NC follows from the dynamics of squared error minimization on the kernel.

**Strengths:**

The paper is extremely well-written and well organized. I did not check the details of the proofs in the appendix but main paper is correct as far I can verify. The relevant literature is well covered in the paper and even more so in the appendix. The invariance derived to establish the connection can be an interesting result on its own and its cooccurrence with NC is demonstrated in realistic experiments.

**Weaknesses:**

While I enjoyed reading the paper, I cannot come to agree with the main premise of the work and so I'm voting for rejection. The results rely on a perfect block structure which is unlikely to be achievable in a realistic setting. Even in a restricted setting where such structure will emerge, it will likely emerge along with NC at the end of training and it is hard to argue that it is a cause or driving force behind NC. I will elaborate on these comments (W1, W2) as well as other weaknesses. Overall, a block NTK would have been interesting as a theoretical model (like unconstrained features) and the derivation of NC from this model is insightful, but the current paper is pushing this model far beyond its domain of applicability by assuming that the NTK of the whole network has this structure.

W1: Perfect block structure in NTK assumes the gradients (of an output) are perfectly orthogonal across data points from different classes. I think a necessary requirement for this would be that the data is already linearly separable at the input or the first hidden layer. Otherwise there is some inevitable interference between the classes in the first few layers where data is not linearly separable, making gradients correlated in those layers and preventing the block structure, no matter how long the network is trained. The paper demonstrates a nearly perfect block structure on the extremely simple task of MNIST and the extremely overparametrized network of ResNet20. The perfect block structure is likely far from realistic in larger tasks and smaller networks.

W2: If the block structure was a property of the NTK at initialization then one could argue for a causal relationship from the block structure to NC. The problem is, both the structure and NC are emergent properties of training dynamics and it is not clear to me why the former would be causing the latter. Previous work arguing that NTK alignment facilitates optimization focuses on alignment already present at initialization or on a partial increase in alignment within an early phase of training. It may be reasonable then to argue that the rest of the training will be faster compared to training with a misaligned NTK. The perfect block structure in this paper is not established as an early phase phenomenon.

W3: I'm surprised by the block structure in the individual hidden layer neurons in Fig 1 (b,d). While the output layer neurons are individually fit to the labels, the hidden layer neurons are updated as a whole through backpropagation. Emergent properties in this layer are then likely to be rotation invariant in general. The fact that each individual neuron is showing a pattern here could be a byproduct of the activation function (ReLU). Does the same pattern emerge with leakyrelu activations (with default parameters)? If not, the current results are limited to ReLU. (I may be mistaken here and perhaps the property is rotation invariant.)

W4: The main result assumes centralized activations and the argument is that Batch Normalization (BN) to some extent enforces this property. Are the representations in the experiments extracted right after a BN layer or right after a ReLU layer? The former would partially satisfy this assumption but the latter would in fact ensure that this is not the case. The text refers to Fig 3 in the appendix but some of the numbers in that figure are ~10. Even if they were smaller, I'm not sure how they would support this assumption since it seems to me that it is not the absolute magnitude of the mean but its magnitude relative to the overall norm of the features that matters.

W5: Previous work on NTK alignment (Baratin et al, Atanasov et al) generally shows a final alignment in the approximate range of 0.1 and 0.5 in different networks and tasks. The kernel alignments in the submission are consistently higher. I understand that different experiments in these works would simply result in different alignment values. Nevertheless the large gap suggests to me that there is a key factor (in training setup or centralizing the kernels or another aspect of the experiment) that differs between the submission and previous work which should be spelled out to prevent the reader from drawing premature conclusions across these papers.

---------

**Post-rebuttal:** I read the other reviews and responses. The rebuttals answered my comment about causality (W2) and so I raised the score. In particular, the response clarified that NTK alignment as measured in the submission is similar to previous work and that it is established as an early-time phenomenon (while NC is a late-time phenomenon). The intro should mention this before claiming a causal relationship. Also showing an example of early increase in alignment instead of the current example in Fig. 1 (e) helps with motivating the causal relationship.

Regarding W1: I agree that the theory doesn't assume a perfect block pattern. My issue is what the title of this paper implies. Unless NTK shows a near perfect block pattern like Fig. 1 (a-d) in more general cases, calling its behavior "collapse" is misleading. The revision should either provide results like Fig. 1 (a-d) in more general cases or clarify that this figure is a rare and extreme case.

Other reviewers argued that the new result has little significance beyond what what is known from unconstrained features model. I do not have the expertise to comment on this issue and leave this decision to other reviewers.

**Questions:**

See weaknesses

**Limitations:**

See weaknesses

---

> ### Author Rebuttal · Authors · 2023-08-04
>
> We thank the reviewer for the careful evaluation of our work and for raising their concerns! Below are our responses.
>
> **W1 orthogonality:** We believe that the reviewer may have misunderstood our main assumption on the NTK structure, since we do not assume that "the gradients (of an output) are perfectly orthogonal across data points from different classes". Our Assumption 3.2 states that $\Theta_{k,k}(x,x')=\langle\nabla f_k(x),\nabla f_k(x')\rangle=\gamma_n\in\mathbb R_+$ for any two elements of different classes $x$ and $x'$. The value of $\gamma_n$ is allowed to be non-zero in our assumption, and it is of course non-zero in practice for the reasons that the reviewer mentioned. We would also like to point out that the reviewer's claim that we assume "the NTK of the whole network has [block] structure" is not correct. We only assume the block structure of the NTK in the last two layers. Therefore, the kernels in the earlier layers are allowed to take arbitrary form.
>
> **W1 linear separability:** Neither NTK alignment nor NC implies that the data is linearly separable in earlier layers. Since we only make assumptions on the kernel in the two outer-most layers, there is a lot of flexibility in the inner layers for DNNs to lift the (possibly non-linearly-separable) data into another high-dimensional space, where we can separate them.
>
> **W1 adequacy of the block structure assumption:** While the illustrative Fig.1 only shows the NTK block structure for ResNet20 trained on MNIST, our experiments report the kernel alignment values for three architectures (ResNet, VGG and DenseNet) and three datasets (MNIST, CIFAR-10 and FashionMNIST). The alignment is high in most of our experiments, so the kernels display an approximate block structure in each setting. This shows that our assumption is realistic for the considered dataset-architecture pairs. We note that our experiments intentionally cover DNNs that display NC in previous literature. While strong NTK alignment may not occur in other DNNs (e.g., smaller or less fit to the task), there is no evidence that NC would occur in such DNNs either.
>
> **W2 causality:** We fully agree with the reviewer that we do not show a causal relationship between NTK alignment and NC but explore the connection between the two empirical phenomena. We will make this more clear in the introduction of the revision. Specifically, we will avoid the misleading statement that NTK alignment "leads" to NC. However, there is empirical evidence that high levels of NTK alignment are achieved before the loss decreases to near-zero values (see Figure 3 in [1]), and that the kernel changes most rapidly in the early stages of training (see [1,2]). These observations justify the analysis of the dynamics with block-structured NTK before the terminal phase of training.
>
> **W3 rotation invariance:** Since the traced kernel displayed in Fig.1b is a matrix inner product $\langle\nabla_wh(x),\nabla_wh(x')\rangle=Tr(\nabla_wh(x)\nabla_wh(x')^\top)$, it is of course rotation invariant w.r.t. $h$ (i.e., it does not change if we multiply $h(x)$ by an orthogonal matrix $A$). Since we assume $\Theta^h(x,x')=\nabla_wh(x)\nabla_wh(x')^\top = \kappa\mathbb I_n$, our assumption is also rotation invariant. We thanks the reviewer for this observation and will add a remark about rotation to the revision.
>
> **W4 batch normalization:** We extract $h(x)$ after ReLU in the experiments and we agree with the reviewer that this setup ensures that the global features mean is not exactly equal to zero in practice. However, our experiments show that the global mean is insignificant (at least an order of magnitude smaller) in comparison with the class means displayed in the same figures, which supports our assumption. We also examine what happens if we discard the zero global mean assumption in the discussion after Theorem 5.1, point (4). In this case, the dynamics does not have to converge to perfect NC.
>
> **W5 alignment values:** Atanasov et al [1] report the alignment values on MNIST for 2-layer MLPs, which are much less powerful architectures than we used in our paper. Previous works show that the alignment increases with depth (see Figure 5 in [3], where the alignment on MNIST reaches 0.75 for a 5-layer MLP), and this increase is associated with better performance. Thus, it is expected that our models display better alignment on MNIST. On the other hand, we report alignment around 0.5 for ResNet trained on CIFAR-10  (see Figure 8 in Appendix C), which is more consistent with Atanasov et al [1]. We believe that the numbers in Baratin et al. [4] are lower because they measure the alignment of matrices of size $NC\times NC$ created by concatenating the features $\nabla f_k(X)\in\mathbb R^{N\times P}$ and labels $\mathbf Y_k\in\mathbb R^N$ over $k=1,\dots C$, while all the other works (including ours) measure the alignment of the traced kernel $\sum_k\Theta_{k,k}(X)\in\mathbb R^{N\times N}$. We note that we used standard architectures and training procedures, which are described in detail in Section 6. We also provide the code to check and reproduce our numerical results.
>
> **References**
>
> [1] Atanasov et al. Neural networks as kernel learners: The silent alignment effect. (2021).
>
> [2] Fort et al. Deep learning versus kernel learning: an empirical study of loss landscape geometry and the time evolution of the neural tangent kernel. (2020).
>
> [3] Shan & Bordelon. A theory of neural tangent kernel alignment and its influence on training. (2021).
>
> [4] Baratin et al. Implicit regularization via neural feature alignment. (2021).

---

> > ### Comment · Reviewer_MvSr · 2023-08-12
> > **Discussion**
> >
> > Thank you for the detailed rebuttal.
> >
> > Regarding W1: In Section 2.2, is lowercase **w** the parameters of the whole network or the last two layers? The kernel defined in Eq 2 is then assumed to have a near perfect block structure in the first part of Assumption 3.2. Are the parameters in this kernel w? And is this the whole network parameters or just the last two layers?

---

> > > ### Author Response · Authors · 2023-08-14
> > >
> > > In Section 2.2, $\mathbf{w}$ are the parameters of the whole network. Then the NTK $\Theta_{k,k}(x,x')=\langle\nabla_{\textbf{w}} f_k(x), \nabla_{\textbf{w}} f_k(x')\rangle$ in Eq. 2 is the inner product kernel of the gradients w.r.t. all the parameters of the network. We note that this is the standard definition of the NTK in the literature.
> > >
> > > Note that one can also write the NTK as a sum $\Theta_{k,k}(x,x’)=\sum_{\ell=1}^L \Theta_{k,k}^{\ell}(x,x’)$, where kernels $\Theta_{k,k}^\ell(x,x’):=\langle\nabla_{\textbf{w}^\ell}f_k(x), \nabla_{\textbf{w}^\ell}f_k(x')\rangle$ are the components of the NTK corresponding to parameters $\textbf{w}^\ell$ of each individual layer $\ell=1,\dots,L$.  Then, as we mentioned in the rebuttal, it it clear that our Assumption 3.2 does not imply that each element of this sum has a block structure. We only assume that the whole sum has an approximate block structure (with non-diagonal blocks *not* equal to zero). Therefore, for Assumption 3.2 to hold, it is enough that only some of the summands have an approximate block structure, and all the other summands can have another structure (e.g. approximately diagonal). Therefore, our assumption does not imply that that the kernel has a block structure in the earlier layers or that the earlier layers of the network can separate the classes.
> > >
> > > As we mentioned in the rebuttal, the approximate block structure of the NTK (as defined in Eq. 2) is also well supported by our numerical experiments. Moreover, earlier works on the NTK alignment study alignment of individual layers and confirm that in practice not all the layers align to the target function equally well [1,2].
> > >
> > > **References**
> > >
> > > [1] Baratin et al. Implicit regularization via neural feature alignment. (2021).
> > >
> > > [2] Lou et all. Feature learning and signal propagation in deep neural networks. ICML (2022).

---

### Official Review · Reviewer_UoyH · 2023-07-04

**Soundness:** 4 excellent
**Presentation:** 2 fair
**Contribution:** 3 good
**Rating:** 5
**Confidence:** 2

**Summary:**

The paper proposes a mechanism behind the empirical phenomenon of Neural Collapse in deep neural networks. The paper derives and analyzes the training dynamics of DNNs with MSE loss and block-structured NTK, identifying three distinct convergence rates in the dynamics.

**Strengths:**

The strengths of the paper include its theoretical rigor, the clarity of its presentation, and the large-scale numerical experiments that support the theory. The paper also provides a new perspective on the empirical phenomenon of Neural Collapse and identifies the conditions under which it occurs.

**Weaknesses:**

The weaknesses of the paper include the assumption of balanced datasets and the lack of exploration of the effects of non-balanced datasets on the dynamics of DNNs with block-structured NTK. The paper also does not explore the effects of adding stochasticity to the dynamics considered in the paper.

**Questions:**

Please dicuss the relationship of your work with this phenomenon [cite1]

[cite 1] Liu D, Wang S, Ren J, et al. Trap of feature diversity in the learning of mlps[J]. arXiv preprint arXiv:2112.00980, 2021.

**Limitations:**

The limitations of the paper include the focus on MSE loss and block-structured NTK, which may not be applicable to other loss functions and NTK structures. The paper also does not provide practical solutions to prevent or mitigate Neural Collapse in deep neural networks.

---

> ### Author Rebuttal · Authors · 2023-08-05
>
> We thank the reviewer for the positive evaluation of our work! Below are our responses to the reviewer's questions and concerns.
>
> **Unbalanced datasets and stochasticity:** Although we do consider only balanced datasets, we believe that our analysis could in principle be generalized for unbalanced data. This would amount to considering a block-structured NTK with blocks of different sizes. Since such generalization would make the calculations more cumbersome, we propose this as one of the future work directions in our paper. We also refer adding stochasticity to the dynamics to future work. As we note in the response to Reviewer LnGq about a possible weakened kernel assumption, we believe that adding centered noise to the kernel in our model should not significantly change the dynamics of features class means. We would also like to point out that, although our theory does not include noise, the experiments certainly include some stochasticity.
>
> **Relationship to [cite1]:** We looked into the work by Liu et al. [cite1], which studies the two-phase phenomenon in the training of MLPs, where feature diversity decreases in the first phase and then increases in the second phase. We believe that this phenomenon may be related to the dynamics of NTK alignment during training. Indeed, increasing similarity between features gradients in the first phase could mean that the NTK values first become more similar across the whole dataset, and only afterwards the block structure emerges. Thus, a possible future work direction is to study and compare the NTK structure in the first and second phases of training.

---

> > ### Comment · Reviewer_UoyH · 2023-08-15
> > **Thanks for your response**
> >
> > I appreciate your feedback. I would keep my current score regarding the response.

---

### Official Review · Reviewer_Hrgu · 2023-07-05

**Soundness:** 4 excellent
**Presentation:** 4 excellent
**Contribution:** 4 excellent
**Rating:** 7
**Confidence:** 4

**Summary:**

The main contribution of the paper "Neural (Tangent Kernel) Collapse" is the connection of the Neural Tangent Kernel (NTK) alignment and Neural Collapse (NC) phenomenon in deep neural networks (DNNs). The authors assume that the empirical NTK develops a block structure aligned with the class labels. They derive the dynamics of DNNs trained with mean squared (MSE) loss and identify three different convergence rates for certain components of the error. They also identify a hyperbolic invariant that captures the essence of the dynamics and use it to prove the emergence of NC in DNNs with block-structured NTK. Also, it provides large-scale numerical experiments to support their theory. Overall, the paper provides valuable insights into the dynamics of DNNs with block-structured NTK and the emergence of NC.

**Strengths:**

- Originality: Yes, this is the first work to connect NTK alignment and NC. Another new contribution is exploring the dynamics of DNNs with block-structured Neural Tangent Kernel (NTK).

- Quality: High quality, the paper presents a thorough analysis of the dynamics of DNNs with block-structured Neural Tangent Kernel (NTK) and the emergence of Neural Collapse (NC) phenomenon. Also, they provide large-scale numerical experiments on three common DNN architectures and three benchmark datasets to support their theory.

- Clarity: Yes, the paper is well-structured and clearly explains technical terms and concepts, making it easy to understand for readers.

- Significance: Yes, for explaining the emergence of NC, most theoretical works adopt the unconstrained feature models. However, this paper provides a new point of view of NTK to explain it. It makes a step towards realistic DNN dynamics by means of the NTK.



**Weaknesses:**

The main weakness of the paper is that its fundamental assumption 3.2 is not justified enough. It assumes the block structure of the NTK kernel but it’s often not the case in real DNNs which is not well-trained.

Some minor remarks:
1. I think the subtitles for a) and b) in Figure 1 should be the **sum** over the classes or feature dimensions.
2. Equation 2: missing part of the parentheses.

**Questions:**

1. This paper only focuses on DNNs trained with mean squared error (MSE) loss. Since Cross Entropy(CE) loss is a common choice for training classification networks, does this dynamics of DNNs and conclusions in this paper also hold for CE loss?

2. Can we get some theoretical insights from this work to improve the design and training of deep neural networks?

**Limitations:**

The authors have addressed their limitations in chapter 7. There is no negative societal impact to be expected from this work

---

> ### Author Rebuttal · Authors · 2023-08-05
>
> We thank the reviewer for the positive evaluation of our work! Below are our responses to the questions.
>
> **Cross Entropy loss:** Generalizing our theoretical results to CE loss is challenging, since the dynamics equations with CE loss are more complex than in case of MSE even with block-structured NTK. In general, CE loss is difficult to analyze, so theoretical NC papers typically focus on unconstrained features models dynamics with MSE loss [1,2]. To the best of our knowledge, NC papers that consider CE only analyze global minimizers of the loss function and do not study dynamics equations [3,4,5]. However, we agree that it is important to verify whether our conclusions also hold for CE loss, and will provide additional empirical results for CE loss in the appendix of the revision.
>
> **Insights for design and training of DNNs:** While not all DNNs display NC, little is known about particular factors that determine whether a given DNN would converge to NC or not. Our work may shed some light on the importance of weights decay and batch normalization for the emergence of NC, which previous literature also conjectured (see [1,6]). In particular, our assumption on the invariant (which is necessary for the emergence of NC) has effects similar to regularization (see e.g. Appendix A.1 in [7]), while the zero global mean assumption is related to batch normalization.
>
> **References**
>
> [1] Han et al. Neural collapse under mse loss: Proximity to and dynamics on the central path. ICLR (2022).
>
> [2] Mixon et al. Neural collapse with unconstrained features. (2020).
>
> [3] Lu \& Steinerberger. Neural collapse with cross-entropy loss (2020).
>
> [4] Zhu et al. A geometric analysis of neural collapse with unconstrained features. NeurIPS (2021).
>
> [5] Wojtowytsch et al. On the emergence of simplex symmetry in the final and penultimate layers of neural network classifiers. (2020).
>
> [6] Ergen \& Pilanci. Revealing the structure of deep neural networks via convex duality. ICML (2021).
>
> [7] Tirer \& Bruna. Extended unconstrained features model for exploring deep neural collapse. ICML (2022).

---

> > ### Comment · Reviewer_Hrgu · 2023-08-18
> >
> > The work is worth admiring. Also, the authors defend themselves well.

---

> > > ### Author Response · Authors · 2023-08-21
> > >
> > > We thank the reviewer again for their valuable feedback and the generous score!

---

### Official Review · Reviewer_LnGq · 2023-07-05

**Soundness:** 3 good
**Presentation:** 4 excellent
**Contribution:** 3 good
**Rating:** 6
**Confidence:** 4

**Summary:**

This work provides a theoretical connection between two related phenomena in deep learning dynamics: the structural change to the empirical NTK during training, specifically its alignment with class labels; and the neural collapse, which refers to a set of behaviors exhibited by NNs trained on multiway classification. The authors asked whether the first phenomenon causes the second, and studied NN dynamics under a toy-model NTK that has a block structure.

**Strengths:**

Both neural collapse and NTK alignment are prominent empirical features of DNN learning (at least in some cases), and the authors provided a careful treatment of a relevant toy model that connects the two. The writing is clear despite the convoluted nature of the subject (speaking as someone who has written on this specific topic before). Assumptions, claims and proven results are easy to find and understand.

**Weaknesses:**

My primary concern is with the amount of insight that this work brings.

(1) The central question posed by the authors, in line 37, is stated as "does NTK alignment lead to neural collapse?". But I do not understand the cause-and-effect relation implied here. It has been long understood that NTK provides a dual perspective to NN training dynamics. So much as the NTK does not "lead to" the reduction of loss in NN training, NTK alignment does not "lead to" neural collapse.

(2) My second point is closely related to the first one. The assumptions about the NTK kernel and feature kernel in Sec. 2.2 are motivated by empirical observations of trained NNs and make ensuing analysis simpler. But I'm concerned that they are sufficiently strong to "guarantee" neural collapse -- these are very strong assumptions and already imply that the NN representation/dynamics are in a stage where things are "collapse-y". In other words, on a conceptual level, this work risks falling into a circular logic -- if we assume the NN is in neural collapse, we can derive that it is in neural collapse. This work does not touch on the topic of how the kind of block-like structure arises, which in my view is what "causes" neural collapse (Conceptually, I think the emergence of block-like kernels and NC are two sides of the same coin).

(3) As the authors pointed out, an important implication of NC is the generalization benefits that it brings. The current analysis, however, appears to entirely focus on the training set. This is a limitation shared by previous work on empirical NTK dynamics, though.

(4) A minor point, but I find Sec. 4.1 to be a little excessive. It is well known that if you do regression with a kernel, eigenstructure of the kernel determines how quickly different components of the target vector gets learned. There are plenty of prior work on the subject. Perhaps it is good to cite them and shorten this section.

**Questions:**

I wonder how much the analysis can be generalized to weakened assumptions about the kernels? For example, instead of assuming that the kernel has a block-like structure, can we assume that it has the form K=K0 + Kb, where Kb is the block-like structure proposed by the authors, and K0 is the eNTK at initialization? I think this is a much more realistic assumption about the dynamics of NN learning.

I also welcome additional comments from the authors about the insights (of course, what counts as "insights" is different for different people!) that this work brings about the emergence of block structures / NC.

**Limitations:**

Please see weaknesses. There are no negative societal impact. I appreciate the author discussing the weakness of making the assumptions in 3.2

---

> ### Author Rebuttal · Authors · 2023-08-04
>
> We thank the reviewer for the careful evaluation of our work! Below are our responses to the reviewer's questions and concerns.
>
> **Does NTK alignment cause NC?** We fully agree with the reviewer that we do not show a causal relationship between NTK alignment and NC but rather explore the connection between the two empirical phenomena. We will make this more clear in the introduction of the revision. Specifically, we will avoid the misleading statement that NTK alignment "leads" to NC. However, there is empirical evidence that high levels of NTK alignment are achieved before the loss decreases to near-zero values (see Figure 3 in [1]), and that the kernel changes most rapidly in the early stages of training (see [1,2]). These observations justify the analysis of the dynamics with block-structured NTK before the terminal phase of training. Combined with these observations, our results also suggest that we could use NTK alignment (together with the invariant identified in our paper) to ``predict'' NC before convergence.
>
> **Are NTK alignment and NC two sides of the same coin?** Our results indicate that NC and NTK alignment are related but are not the same thing. Indeed, we show (both theoretically and empirically) that DNNs with NTK alignment do not always converge to NC. Moreover, our assumption on the dynamics invariant provides a necessary condition for the emergence of NC in DNNs with block-structured NTK (see Theorem 5.1 and the following discussion). Thus, we believe that NTK alignment is a more common phenomenon than NC, and this insight is one of the main contributions of our work.
>
> **Does NC follow trivially from the NTK block structure?** While the connection between NTK alignment and NC may seem obvious to some readers on the intuitive level, establishing such connection theoretically is not entirely trivial and, to the best of our knowledge, it has never been done before in the literature.
> In the response to Reviewer PDBS, we explain why NC does not follow trivially from the assumption on the NTK block structure. Moreover, as we mentioned above, NC does not even always occur in DNNs with block-structured NTK.
>
> **Broad implications:** Our work proposes to study NC through the lens of NTK alignment, which opens new research directions. Previous works on NC focus on the top-down approach (layer-peeled models) and fundamentally cannot explain how NC develops through earlier layers of a DNN, what are the effects of depth, etc. On the other hand, NTK alignment literature focuses on the alignment of individual layers, and recent theoretical works even quantify the role of each hidden layer in the alignment [3]. Therefore, we believe that the connection between NTK alignment and NC established in our work provides a conceptually new method to study NC. We will include a discussion section in the revision to explore the implications of our work in more detail.
>
> **Weakened kernel assumption:** The NTK at initialization is random (w.r.t the random initialization parameters) and has an approximately diagonal structure (see [4,5]). Therefore, a kernel of the form $\Theta = \Theta_{block} + \Theta_0$ can be modeled as a certain block-structured kernel $\Theta_{block} := a\mathbb{I} + b\mathbf Y^\top\mathbf Y + c\mathbf{1}\mathbf{1}^\top$ plus an i.i.d. noise term. Under this model, the dynamics of the features' class means $\langle h \rangle_c$ can be viewed as an approximation of the expected dynamics for each class. Thus, assuming that the noise is centered and the sample size is large enough, the noise should not change the dynamics of the class means too much. However, quantification of the randomness in such a setting may be theoretically challenging and we propose it as a future work direction in our paper.
>
> **References**
>
> [1] Atanasov et al. Neural networks as kernel learners: The silent alignment effect. (2021).
>
> [2] Fort et al. Deep learning versus kernel learning: an empirical study of loss landscape geometry and the time evolution of the neural tangent kernel. NeurIPS (2020).
>
> [3] Lou et all. Feature learning and signal propagation in deep neural networks. ICML (2022).
>
> [4] Xiao et al. Disentangling direct from indirect relationships in association networks. (2022).
>
> [5] Seleznova et al. Neural tangent kernel beyond the infinite-width limit: Effects of depth and initialization. ICML (2022).

---

> > ### Comment · Reviewer_LnGq · 2023-08-18
> > **Thank you**
> >
> > Thanks to the authors for their careful response. The response and other proposed edits would address my concerns significantly. I am therefore raising my score.

---

> > > ### Author Response · Authors · 2023-08-21
> > >
> > > We are grateful to the reviewer for taking our arguments into account and increasing the score! We also greatly appreciate the reviewer's feedback, which helped us to convey the results and implications of our work more clearly.

---

### Official Review · Reviewer_PDBS · 2023-07-25

**Soundness:** 2 fair
**Presentation:** 3 good
**Contribution:** 2 fair
**Rating:** 4
**Confidence:** 4

**Summary:**

This paper connects NTK (neural tangent kernel) and NC (neural collapse) by assuming NTK has a block structure in the late training stage, meaning the kernel for samples within the same class is much larger than samples from different classes. The technical assumption additionally assumes that the gradients of classification heads are independent of each other, and the gradients of neurons of the penultimate layer are also independent of each other. The authors then claim that the kernel gradient descent of this NTK leads to NC. Empirical results provide the correlation between the block structure and NC.

**Strengths:**

The paper is the first paper I have seen that builds the connection between NTK and NC. It is inspiring in concept with good novelty.

The presentation of the paper is overall good and easy to read.

**Weaknesses:**

I believe the assumption of the paper is too strong because the authors not only assume the block structure of NTK but also assume the independence between gradients of neurons/classification heads for any input. Taking a closer look at Assumption 3.2, one can already derive strong results. For example, one can wrote $\nabla f_k(x)$ as a function of $\nabla h_s(x)$, $\mathbf W$ and $h_s(x)$ by chain rule, and plug into $\Theta$. Then it is not hard to find that

-  $\langle \mathbf W_k, \mathbf W_s\rangle$ is identical for any $k\neq s$;
- $\langle \mathbf W_k, \mathbf W_k-\mathbf W_s\rangle=\gamma_d/\kappa_d=\gamma_c/\kappa_c=\gamma_n/\kappa_n$;
- $\langle h(x), h(x')\rangle =-\kappa \langle \mathbf W_k, \mathbf W_s\rangle$ ($\kappa$ is $\kappa_d, \kappa_c$ or $\kappa_n$ depends on the relationship between $x$ and $x'$).

Those are some implications I derived within an hour. There might be some flaws but the message is clear --- the independence between gradients is a very very strong assumption.

Another thing I want to point out in the proof is that the training may not be in the kernel regime. The authors assume after hundreds of training epochs, the empirical NTK satisfied the property, but it is not clear whether it changes afterward (meaning it is not in the kernel regime but feature learning regime, which does not support linearization of the model).

**Questions:**

Can you verify the calculation I did above?

**Limitations:**

I do not see any potential negative societal impact.

---

> ### Author Rebuttal · Authors · 2023-08-04
>
> We thank the reviewer for rising the concern about the assumption of the independence between output/feature neurons, i.e., the part of Assumption 3.2 stating that $\Theta_{k,k'}(x,x')=\Theta^h_{k,k'}(x,x')=0$ for any $k\neq k'$ and any $x,x'$. While the calculations in the review do not appear fully correct (we provide details below), we agree that the independence assumption is quite strong. In fact, as we show below, we can get rid of the independence part of the assumption with minimal changes to the paper.
>
> **Relaxation of the assumption on $\Theta^h$:** Based on the review, we identified that we can relax our main assumption without significant changes to the paper in the following way:
> $$\Theta^h_{k,k'}(x,x')=\beta\cdot\Theta^h_{k,k}(x,x')\quad \forall k=1,\dots,n,\forall k'\neq k $$
> for some $0\leq\beta<1$. Note that our original assumption is the special case of this assumption with $\beta=0$. Instead of requiring independence between different features, the relaxed assumption only states that the dependence between different features is weak, which is indicated by $\beta$.
>
> This relaxation amounts to a simple change in the training dynamics of DNNs with block-structured NTK. Indeed, in the proof of Theorem 4.1 (Appendix B.1), we obtain the following dynamics for $h_s(x_i)$:
> $$ \dot h_s(x_i)=-\sum_{s'=1}^n\sum_{i'=1}^N[ \mathbf W\mathbf H-  \mathbf b\mathbf 1_N^\top-\mathbf Y ]\_{s'i'}\Theta^h_{s',s} (x_i,x_{i'})$$
> Vectorizing each term of the outer sum, we get the dynamics of the whole features matrix:
> $$\dot{\mathbf H}=-\mathbf A[\mathbf W^\top(\mathbf W\mathbf H+\mathbf b\mathbf 1_N^\top-\mathbf Y)][(\kappa_d-\kappa_c)\mathbb I_N+(\kappa_c-\kappa_d)\mathbf Y^\top\mathbf Y + \kappa_n\mathbf 1_N\mathbf 1_N^\top],$$
> where $$\mathbf{A}:=(1-\beta)\mathbb I_n + \beta\mathbf 1_n\mathbf 1_n^\top.$$
>
> Thus, adding the dependence between the feature neurons amounts to "scaling" the dynamics of $\mathbf H$ by an invertible matrix $\mathbf A$. The dynamics of $\mathbf W$ and $\mathbf b$ remain unchanged.
>
> We notice that such scaling does not effect our proof of variability collapse (NC1). Indeed, applying a change of variables $\tilde{\mathbf E}:=\sqrt{\mathbf A}{\mathbf E}\sqrt{\mathbf A}$ and $\tilde{\mathbf H}_2:=\sqrt{\mathbf A^{-1}}\mathbf H_2$, we can carry on the same proof and show that $\tilde{\mathbf H}_2\to\mathbb O$ and thus $\mathbf H_2\to\mathbb O$. Similarly, we can apply a change of variables $\tilde{\mathbf W}:=\mathbf W\sqrt{\mathbf A}$ and $\tilde{\mathbf H}_1:=\sqrt{\mathbf A^{-1}}\mathbf H_1$ in the proofs of NC2-4 to show the duality $\tilde{\mathbf H}_1\propto\tilde{\mathbf W}^\top$. From this duality and the assumptions of Theorem 5.1, we get the ETF structure of the features matrix and the duality $\mathbf H_1\propto\mathbf W^\top$. Thus, the statement of the main Theorem 5.1 remains unchanged under the relaxed assumption.
>
> We will adopt the relaxed version of the assumption in the revision of our paper.
>
> **Relaxation of the assumption on $\Theta$:** It is also possible to relax the assumption on the NTK $\Theta$ in the same way:
> $$\Theta_{k,k'}(x,x')=\beta'\cdot\Theta_{k,k}(x,x')\quad\forall k, \forall k'\neq k$$
> for some $0\leq\beta'<1$. This relaxation does not effect the dynamics derived in Theorem 4.1, which is the main target of our analysis. The assumption on $\Theta$, in fact, only effects the convergence analysis in Section 4.1. The argument about the convergence rates still holds with the relaxed assumption, since the eigenvectors of the non-diagonal terms of the NTK $\Theta_{k,k'}$ are the same as of the diagonal terms $\Theta_{k,k}$.
>
> **Implications of the assumption:** Following the reviewer's reasoning and writing $\nabla f_k(x)$ by chain rule as a function of $\nabla h(x)$, $\mathbf W$ and $h(x)$, we can derive the following two implications of our relaxed assumption:
> \begin{align}
>     \langle\mathbf W_k,\mathbf W_{k'}\rangle&=-\beta\sum_{s\neq s'}\mathbf W_{ks}\mathbf W_{k's'}+\dfrac{\Theta_{k,k'}(x,x')}{\Theta^h_{s,s}(x,x')}\quad\forall k\neq k',\\\\
>     \langle\mathbf W_k,\mathbf W_{k}\rangle &=-\beta\sum_{s\neq s'}\mathbf W_{ks}\mathbf W_{ks'}+\dfrac{\Theta_{k,k}(x,x')}{\Theta^h_{s,s}(x,x')}+\dfrac{\langle h(x), h(x') \rangle+1}{\Theta^h_{s,s}(x,x')}
> \end{align}
> As the reviewer correctly noted, this implies $\langle\mathbf W_k,\mathbf W_{k'}\rangle=const$ for any $k\neq k'$ in the special case of $\beta=0$ (but not if $\beta>0$). The two remaining conclusions in the review do not seem to hold even in case $\beta=0$. In particular, as far as we can see, neither the ETF structure of the class means (NC2), nor the duality between the weights and the features matrix (NC3) follow from these equations. Moreover, even variability collapse (NC1) does not in general follow from the assumption.
> A trivial counterexample where $\mathbf H$ satisfies the above conditions but NC1-2 do not hold is given by the following configuration of the feature vectors: $h(x_1^{c_1})=(1,0,0,0)$, $h(x_2^{c_1})=(1/\sqrt{2},1/\sqrt{2},0)$, $h(x_1^{c_2}) = (0,0,0,1)$, $h(x_2^{c_2}) = (0,0,1/\sqrt{2},1/\sqrt{2})$.
>
> **Kernel regime:** There is empirical evidence that high levels of NTK alignment are achieved before the loss decreases to near-zero values (see Figure 3 in [1]), and that the kernel changes most rapidly in the early stages of training (see [1,2]). These observations justify the analysis of the dynamics with block-structured NTK before the terminal phase of training. Nevertheless, we of course do not claim that the NTK of real DNNs remains completely constant even in the end of training. Generalizing the analysis to non-constant NTK is an interesting but challenging future work direction.
>
> **References**
>
> [1] Atanasov et al. Neural networks as kernel learners: The silent alignment effect. (2021).
>
> [2] Fort et al. Deep learning versus kernel learning: an empirical study of loss landscape geometry and the time evolution of the neural tangent kernel. NeurIPS (2020).

---

> > ### Comment · Reviewer_PDBS · 2023-08-21
> > **Comments on the implication and relaxation**
> >
> > Thanks to the authors for the clarification! My quick calculation indeed had a bug, so my second and third implications are false. I also verify the two implications derived by the authors. However, if we take a closer look at the correct implications, we can find that, without the newly proposed relaxation,
> > - $\langle W_k,W_{k'}\rangle=0$.
> > - $\langle W_k, W_k\rangle=\gamma/\kappa+(\langle h(x), h(x')\rangle +1)/\kappa$ where $\gamma,\kappa$ depends on the relation between $x,x'$.
> >
> > They are very strong assumptions. Even with the relaxation, given the relation between $x$ and $x'$, $\langle h(x), h(x')\rangle$ is still a fixed value.
> > Overall, I think the assumption (even with relaxation) is strong because of too much symmetry, which is convenient for proving NC.
> >
> > On the other hand, although I understand the changes in the proof might be straightforward when incorporating the relaxation, it is very hard for me to verify it confidently without reading it with all the changes added.

---

> > > ### Author Response · Authors · 2023-08-21
> > >
> > > We thank the reviewer for taking the time to verify our calculations and getting back to us! We would like to highlight the following points that we made in the rebuttal regarding our assumption:
> > >
> > > - The NTK block structure assumption alone does not imply NC. Moreover, none of the four NC components (denoted NC1-4 in the paper) follows from the assumption alone. We show this 1) in the discussion after the main Theorem 5.1, and 2) in the counterexample on $\mathbf{H}$ in the rebuttal ("Implications of the assumption"). Therefore, even perfect NTK block structure does not guarantee NC, and our work provides necessary conditions for the emergence of NC in DNNs with NTK alignment. These necessary conditions cannot be derived directly from the assumption and require analysis of the dynamics presented in the paper.
> > > - The NTK block structure assumption is supported by a large body of empirical evidence (including our numerical experiments and previous works on NTK alignment). Hence, we believe that our assumption is a justified simplification of realistic DNNs' behaviour.
> > >
> > > Therefore, while our assumption certainly makes the analysis of the DNNs dynamics much simpler, our results are still 1) non-trivial, 2) approximate behaviour of realistic DNNs, 3) provide new insights into NC (see also the response to Reviewer LnGq regarding the broad implications of our results). Moreover, since our results are novel and connect two empirical phenomena widely studied by separate sections of ML community, we believe that our work may be interesting and potentially insightful for a wide audience.

---

> > > > ### Comment · Reviewer_PDBS · 2023-08-21
> > > >
> > > > Thank the authors for replying. Regarding your highlights, I agree that your assumption does not imply NC, but I believe your assumption is not fully justified by empirical evidence. There are certainly additional structures in Figure 1 that are not captured by your assumption. (BTW, you should probably add figure legends in Figure 1 to indicate the color-value mapping, especially what color corresponds to 0.)
> > > >
> > > > I really like your idea of connecting NC with NTK, but I am afraid the contribution is not significant enough for NeurIPS because of the strong assumptions.

---

> > > > > ### Author Response · Authors · 2023-08-21
> > > > >
> > > > > We thank the reviewer again for the quick response. We would like to note that the only (to the best of our knowledge) previous work connecting NC and local elasticity of neural networks [1] was in fact published in NeurIPS. We discuss this work in Appendix A ("Relevant Works»), paragraph «NC and local elasticity». The authors of [1] model local elasticity of the training dynamics through a *block-structured effect matrix*, which has only two distinct values: a larger intra-class value and a smaller inter-class value. However, the authors do not provide any explicit connection between the effect matrix and real gradient flow dynamics of DNNs but simply assume that the local elasticity values affect the dynamics of the output/features linearly.
> > > > >
> > > > > On the other hand, our assumption explicitly describes how local elasticity emerges from the NTK structure, and our dynamics is derived from the gradient flow equations. Therefore, we believe that the assumption made in our work is not stronger than in relevant works presented at NeurIPS.
> > > > >
> > > > > [1] Zhang, J., Wang, H., & Su, W. (2021). Imitating deep learning dynamics via locally elastic stochastic differential equations. Advances in Neural Information Processing Systems, 34, 6392-6403.

---

### Author Rebuttal · Authors · 2023-08-05

We thank all the reviewers for their valuable feedback! Based on the reviews, we were able to identify several improvements for the paper, which we will incorporate in the revision. Below we summarize the reviewers' concerns, our responses, and the proposed changes to our paper.

## Concerns about the NTK block-structure assumption
Reviewers PDBS, LnGq and MvSr expressed several concerns about Assumption 3.2 on the NTK block-structure. We summarize these concerns and our responses below:

- **Ortogonality of gradients between different classes:** Reviewer MvSr states that our assumption implies that "gradients (of an output) are perfectly orthogonal across data points from different classes". However, as we state in the response, we do not assume orthogonality between points from different classes, since $\Theta_{k,k}(x,x')=\gamma_n\geq 0$ in Assumption 3.2. Therefore, we believe that this concern comes from a misunderstanding of our main assumption.
- **Independence between gradients of different feature neurons:** Reviewer PDBS states that the assumption of the independence between different feature neurons is too strong, since it already implies strong results related to NC. Our response is two-fold: 1) We show that we can relax our Assumption 3.2 to remove the independence. Such relaxation does not change the paper's main Theorem 5.1 and requires only small adjustments of the proofs (change of variables). 2) We show that NC does not follow from Assumption 3.2.
- **NC follows trivially from the assumption:** Reviewer LnGq expressed a related concern that our assumption is "sufficiently strong to "guarantee" NC". However, our results indicate that NTK block structure does not guarantee convergence to NC, and our invariant assumption provides a necessary condition for the emergence of NC in DNNs with block-structured NTK. We also detail why NC does not follow from the assumption in the response to Reviewer PDBS.
- **Adequacy of the assumption:** Reviewer MvSr expressed a concern that block structure of the NTK is not realistic and implies linear separability of data in the earlier layers. In the response, we explain that linear separability in earlier layers does not follow from our assumption. Moreover, our numerical results and previous works on NTK alignment provide a large body of empirical evidence to justify our assumption. We believe that this concern may come from a misunderstanding of our assumption and its implications.


We propose the following changes in the revision of our paper to address the reviewers' comments and concerns:

- Based on the comments by Reviewer PDBS, we will adopt a relaxed version of Assumption 3.2 without independence of different features neurons. This improvement requires minimal changes in the paper's reasoning and proofs.
- Based on the comments by Reviewer LnGq, we will modify the introduction to make it clear that NTK alignment and NC are not the same phenomenon and do not always occur together.

Overall, we admit that our main assumption is a simplification of the realistic DNNs’ behavior. However, empirical evidence (including our numerical experiments) shows that our assumption can approximate well-trained realistic DNNs. Moreover, much like our assumption, perfect NC is also a simplification that never holds exactly in realistic DNNs.

## Concerns about significance
Reviewers LnGq and MvSr expressed concerns about significance and implications of our work. We summarize these concerns and our responses below:

- **Causality:** Reviewers LnGq and MvSr share a concern about the causality between NTK alignment and NC. We agree with the reviewers that we do not show a causal relationship between NTK alignment and NC but rather explore the connection between the two empirical phenomena. However, we also reference empirical evidence that high levels of NTK alignment are achieved before the loss decreases to near-zero values, which justifies the analysis of the dynamics with block-structured NTK before the terminal phase of training.
- **Insights:** The primary concern of Reviewer LnGq is the amount of insight that our work brings. We respond that our results are 1) novel, 2) non-trivial, 3) open new research directions. Therefore, we believe that they are valuable for the community.

We propose the following changes in the revision of our paper to address the reviewers' comments and concerns:

- We will modify the introduction to make it clear that we do not show causality between NTK alignment and NC. Specifically, we will avoid the misleading statement that NTK alignment "leads" to NC.
- We will include a new section to discuss implications and broad impact of our work.
- To address the question of Reviewer Hrgu about generalization of our results to CE loss, we will add numerical experiments with CE loss in the appendix of the revision.

Overall, our paper provides the first (to the best of our knowledge) theoretical connection between NTK alignment and NC. Since both phenomena are 1) of high interest for the ML community and 2) hard to study theoretically, we believe that our results would be interesting for a wide audience.

## Conclusion

We believe that the proposed adjustments cover all the major concerns about the paper's technical contribution raised by the reviewers. These adjustments also do not require major changes of the paper's content and proofs. Moreover, while most of the reviewers agree that our results are novel and interesting, we hope that our response made the significance of our work even more clear. We are optimistic that the reviewers will consider our arguments and be open to a partial reevaluation of their scores. We will also gladly respond to any additional questions that may arise during the discussion period!

---

### Decision · Program_Chairs · 2023-09-21

**Decision:**

Accept (poster)

**Comment:**

This paper establishes a connection between the Neural Tangent Kernel (NTK) and Neural Collapse (NC) by assuming that the NTK exhibits a block structure during the late training stage. This structure implies that the kernel values for samples within the same class are significantly larger than those for samples from different classes. The authors prove that the kernel gradient descent based on this NTK leads to the emergence of NC. Empirical results are presented to demonstrate the correlation between the block structure and NC.

The authors' responses have effectively addressed most of the concerns raised by the reviewers. Most reviewers have increased their scores and recommend acceptance, with the exception of one reviewer who appreciates the idea of connecting NC with NTK but expresses concerns regarding the block-structure assumption. As the block structure is approximately observed during training, and the connection between NC and NTK provides additional insights compared with previous approaches based on the unconstrained features model, I recommend acceptance. The authors are advised to address this concern by incorporating the discussions in the responses, along with other comments raised by the reviewers, in the final version of the paper.